# FEDTRANS: CLIENT-TRANSPARENT UTILITY ESTIMATION FOR ROBUST FEDERATED LEARNING

**Mingkun Yang,**[*] **Ran Zhu,**[*] **Qing Wang, Jie Yang**
Department of Software Technology
Delft University of Technology
{m.yang-3,r.zhu-1,qing.wang,j.yang-3}@tudelft.nl

## ABSTRACT

Federated Learning (FL) is an important privacy-preserving learning paradigm that plays an important role in the Intelligent Internet of Things. Training a global model in FL, however, is vulnerable to the data noise across the clients. In this paper, we introduce **FedTrans**, a novel client-transparent client utility estimation method designed to guide client selection for noisy scenarios, mitigating performance degradation problems. To estimate the client utility, we propose a Bayesian framework that models client utility and its relationships with the weight parameters and the performance of local models. We then introduce a variational inference algorithm to effectively infer client utility at the FL server, given only a small amount of auxiliary data. Our evaluation results demonstrate that leveraging FedTrans to select the clients can improve the accuracy performance (up to 7.8%), ensuring the robustness of FL in noisy scenarios [1].

## 1 INTRODUCTION

We live in a world with billions of Internet of Things (IoT) devices deployed to perform various tasks. These devices sense the surroundings and upload the collected data to servers, where techniques such as deep learning are used to perform various tasks such as sensing human activities. The data transfer to cloud or edge servers raises privacy issues. Federated Learning (FL), a distributed training paradigm, has thus been proposed to preserve privacy (McMahan et al., 2017). In FL, a cloud or edge server coordinates the training of local models in connected devices, i.e., clients, to learn a global model. The clients only update the server about the weights of their local models. Based on many rounds of the updates from the clients, FL learns the global model gradually by aggregating clients' weights. Since there is no need for the clients to upload their sensed data, FL has become a popular privacy-preserving learning framework and is being heavily studied for many privacy-sensitive application scenarios such as disease diagnosis (Yang et al., 2021), monitoring (Wu et al., 2020), and object detection (Liu et al., 2020).

A key challenge of FL is the data utility of the involved clients. The widely accepted norm in FL is that the server adopts the random selection strategy and treats each client with the same selection probability. This is based on the assumption that all the clients with heterogeneous distribution can acquire high-quality labels/data and have equal contributions to the model learning. In reality, this assumption does not stand. Studies have reported that existing datasets could easily have more than 30% label errors (Konovalov et al., 2017; Ren et al., 2017). FL server inevitably aggregates the updated local weights from unreliable clients which could degrade the performance of the global model trained at the FL server. The issue is further complicated by the compounded relationship between noisy data and the heterogeneous local data distributions (non-iid), which affects the convergence and final performance of the model. To demonstrate this, we conduct an experiment, using the CIFAR10 dataset and without loss of generality flipping the labels at the clients, with an average label flipping rate of 16.5% across the noisy/corrupted clients, and show the results in Figure 1 (the results under more types of noise can be found in Section 3). We have the following observation: *unreliable clients –the clients with noisy data– slow the learning of the global model and reduce its*

---

[*]Equal contribution
[1]Code is available at https://github.com/Ran-ZHU/FedTrans

*accuracy*. This can be observed clearly in the non-iid setting by comparing the dash-red curve and solid-red curve. Randomly selecting clients under the non-IID setting with/without noise, fails to account for the potential variations in data characteristics across clients, leading to suboptimal performance in terms of the model convergence and the final accuracy. This performance degradation also exists under the IID setting (dash-green curve vs. solid-green curve).

The above observation validates the importance of selecting reliable clients for training the global model at the FL server. To achieve this, a critical step is to assess the client utility in a way that can reflect the quality of updates uploaded by clients in the most practical non-IID scenarios. However, it is nontrivial to infer client utility due to FL server's lack of accessibility to clients' local data. Therefore, the key problem is how to infer client utility without compromising privacy. Ideally, the estimation of the client utility should be transparent to the clients, i.e., no additional operations are required on the client side.

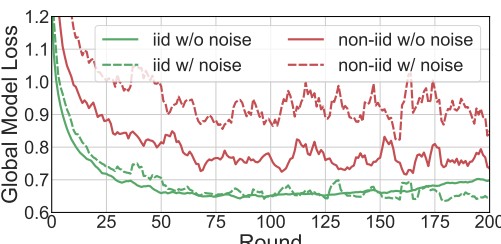

Figure 1: The noise impact on the global model performance shows different patterns under two data distributions illustrating that data heterogeneity and quality are tightly coupled issues.

In this work, we design FedTrans (Figure 2), a novel client utility evaluation framework that enables client selection guidance for heterogeneous and noisy scenarios to ameliorate performance degradation. We exploit a small but high-quality auxiliary data to explore two complementary strategies for client utility inference: 1) creating synthetic clients based on the auxiliary data to build a discriminator that distinguishes high-utility clients from lower ones based on their weight parameters, and 2) evaluating the performance of local models as an indicator of client utility. We propose a unified Bayesian framework that seamlessly couples these two strategies and further introduces a variational inference algorithm that allows the inference based on these strategies to benefit from each other, thereby reaching an effect where the whole is greater than the sum of its parts. We execute experiments on two public datasets under the Dirichlet distribution, using six different label/feature/hybrid noise types. The results demonstrate that our FedTrans delivers better accuracy performance (up to 7.8%) and significantly faster convergence speed (a reduction of over 56% in time consumption, refer to Appendix E) compared to competing FL methods in noisy heterogeneous settings.

## 2 FEDTRANS DESIGN

To overcome the data noise issue, it is important to precisely estimate the utility of parameter updates from clients for updating the global model. For this problem, previous work mainly takes an approach that estimates utility by comparing weight updates among local models and treating updates that disagree more with those from other clients as less useful (Li et al., 2021; Xu et al., 2022). The effectiveness of such methods depends on the validity of two assumptions: 1) that the majority of local updates are high-quality, which is not necessarily always true in real applications; and 2) that data noise is the only factor of the quality of local updates, which largely overlooks the intertwined effect heterogeneous local data distributions bring on the resulting weight updates in local clients. An alternative approach is to infer the update utility by evaluating participating clients against a certain benchmark, by including high-quality auxiliary data with correct labels and a distribution representative of the data distribution that the global model encounters in the application. We note that in the real applications, it is feasible for the service provider to obtain a small amount of auxiliary dataset (Jeong et al., 2018; Tuor et al., 2021; Yang et al., 2022a) (e.g., by soliciting data from paid, anonymous workers). Such an approach does not compromise user privacy provided that the auxiliary data is hosted on the server and the computation is also carried out on the server. A practical limitation of such an approach, however, is that the auxiliary data comes at a cost – as it requires human labor for labeling – and thus the data can only be of limited size. It, therefore, remains an important open question how to best leverage a limited amount of auxiliary data in a cost-effective way for estimating the client's update utility for robust FL.

**Problem Definition.** The $i$-th round of FL involves a set of participating clients $\mathcal{J}^i$ that report their local updates $\{\mathcal{W}^*_{i,j}\}_{j \in \mathcal{J}^i}$ to the server. Given a set of well-labeled and balanced data $\mathcal{D}_a$, the server has to infer the utilities of local updates $\{\theta_j\}_{j \in \mathcal{J}^i}$ to guide the selection of a subset of clients

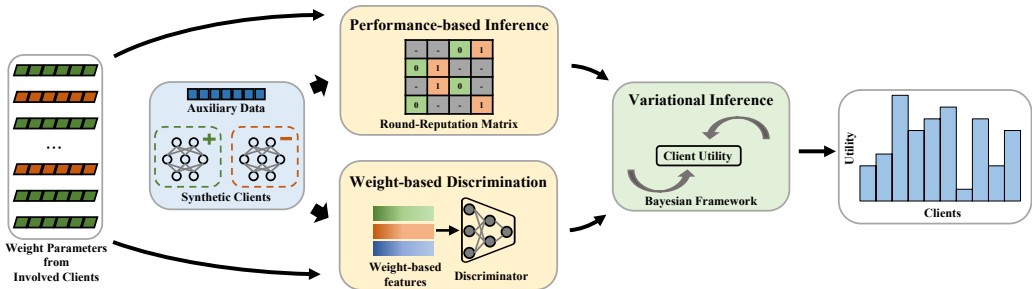

Figure 2: **FedTrans pipeline:** it leverages a small, auxiliary data on the server to infer client utility, using a Bayesian framework that considers both the weight parameters and performance of local models. A learnable discriminator outputs the client utility while the training lacks labels. To tackle this, we apply a variational inference algorithm to update the parameters of the discriminator.

$\hat{\mathcal{J}}^i \subseteq \mathcal{J}^i$ for global model aggregation. The process of utility estimation at the server is *transparent* to all the clients; in other words, no additional operations are required on the client side.

In this way, the model aggregation performed at the server can be formulated as:

$$\bar{\mathcal{W}}_i^* = \arg\min_{\mathcal{W}} \sum_{j \in \hat{\mathcal{J}}^i} \hat{p}_j \mathcal{L}_{\mathcal{D}_j}(\mathcal{W}), \tag{1}$$

where $\bar{\mathcal{W}}_i^*$ refers to the weight parameters of the global model; $\hat{p}_j$ is the weight of the local client $j$ in aggregation satisfying $\sum_{j=1}^{|\hat{\mathcal{J}}^i|} \hat{p}_j = 1$, e.g., the fraction of local samples; $\mathcal{L}_{\mathcal{D}_j}(\cdot)$ is the corresponding training loss. In this paper, we mainly focus on the estimation of client utility; that means, $\hat{p}_j$ and $\mathcal{L}_{\mathcal{D}_j}(\cdot)$ can be arbitrary existing aggregation and local training schemes.

**Privacy Guarantee.** As a client-transparent approach, auxiliary data $\mathcal{D}_a$ resides in the central server. In this way, FedTrans retains the same level of privacy protection as FedAvg or other SOTA frameworks, since the local data and the training process remain unchanged from the user's perspective. An additional advantage of including auxiliary data is that it makes FL more resilient to adversarial attacks: it becomes more challenging to disguise malicious behavior since the performance of the local model can be easily evaluated on the auxiliary data.

## 2.1 MODELING THE CLIENT'S UTILITY

FedTrans makes use of the auxiliary data to infer client utility $\theta_j$ by fusing two kinds of information: 1) the weight parameters from clients and 2) the clients' performance on the auxiliary data. We illustrate the full pipeline in Figure 2 and introduce details in this subsection.

To start, we denote the utility of client $j$ as $\theta_j \in [0, 1]$. $\theta_j$ parameterizes a Bernoulli distribution that generates the selection decision $s_j$ of client $j$:

$$s_j \sim Ber(\theta_j). \tag{2}$$

**Weight-based Utility Estimation.** The auxiliary dataset allows us to simulate a few clients with the known utility using the auxiliary data and based on that, train an additional machine learning model to discriminate clients with low utility by their weight parameters from the rest. To this end, we build a secondary machine learning model, denoted as $f^{\mathcal{W}_d}$, and client utility $\theta_j$ is conditioned on the parameters of local updates, through the weight-based discriminator $f^{\mathcal{W}_d}$:

$$\theta_j = f^{\mathcal{W}_d}(\mathbf{x}_j), \tag{3}$$

where the $\mathbf{x}_j$ refers to the top-layer of local model $\mathcal{W}_{i,j}^*$ in $i$-th round. We specifically consider the weights of the topmost layer of the models from clients, since those weights are most relevant for the given task (Li & Zhan, 2021) and hence most discriminative for utility inference. Note that any DL model for the discriminator can generate $\theta_i \in [0, 1]$ with a sigmoid function.

To train the discriminator, we utilize the auxiliary data $\mathcal{D}_a$ to generate the synthetic clean and corrupted weight parameters, which provides discriminator labeled training samples. Given a set of

clean, well-labeled, and balanced samples $\mathcal{D}_a$, we first construct a clean dataset $\mathcal{D}^+ = \mathcal{D}_a$ and a corrupted dataset $\mathcal{D}^-$ where all the labels are manually flipped to incorrect classes. FedTrans then assigns $\mathcal{D}^+/\mathcal{D}^-$ to $K$ pairs of synthetic positive/negative clients $\{\mathcal{D}_k^+/\mathcal{D}_k^-\}_{k=\{1,\cdots,K\}}$, respectively, where each pair possessed $\lfloor \frac{|\mathcal{D}_a|}{K} \rfloor$ auxiliary samples. For each synthetic client, we obtain synthetic weight parameters by initializing it with the global model from the last round, and then train the model as we would do for normal clients. Training discriminator benefits from such a set of top-layer weight parameters $\{\mathbf{x}_k^+/\mathbf{x}_k^-\}_{k=\{1,\cdots,K\}}$ with known labels.

Weight-based discrimination, however, is intrinsically limited by the size of the auxiliary data, which may only cover a small fraction of the weight patterns of clean and corrupted clients. This is especially a concern given we only consider synthetic clients with all correct or all incorrect labels – in reality, clients may hold labels with a different mixture of correct and incorrect labels.

**Performance-based Utility Estimation.** To address the issue, we also leverage the auxiliary data with correct labels for evaluating the performance of clients and use such performance as an indicator of client utility. To this end, we propose to create a round-reputation matrix $\mathbf{R}$ that keeps track of the local model performance of clients involved in different rounds. In such a matrix, the entry $\mathbf{R}_{i,j}$ in the $i$-th row and $j$-th column records the performance of client $j$ in round $i$. We devise the following way for determining values in the matrix: $\mathbf{R}_{i,j} = 1$ if the performance of the local model from client $j$ exceeds an empirical threshold, and $\mathbf{R}_{i,j} = 0$ otherwise. Note that the threshold should be dynamic with the FL progress, thus we take the performance of the local models (i.e., average accuracy) in $i$-th round on the auxiliary dataset as the threshold.

The round-reputation matrix $\mathbf{R}$ constructed above is a sparse matrix because only a relatively small proportion of candidate clients could be activated in each round. The value assignment for the round-reputation matrix in a certain round is severely affected by the randomness of client involvement, which means that for those $\mathbf{R}_{i,j}$'s that are not blank, the entries cannot be directly treated as the selection decision $s_j$. A more sophisticated way would be to consider the consistency of client performance across different rounds and only consider the clients whose performance consistently exceeds a threshold in multiple rounds as high-utility ones. A further consideration is to account also for the informativeness of different rounds, e.g., client performance in the first few rounds of FL might be less informative due to the randomness of weight initialization in the local models.

We denote round informativeness as $r_i \in [0,1]$; $r_i = 1$ means the $i$-th round provides sufficient information for inferring client utility, $r_i = 0$ otherwise. To account for the uncertainty in estimating $r_i$ and hence to make Fed-Trans more robust, we take a Bayesian view and model the prior probability distribution as a Beta distribution:

$$r_i \sim Beta(\alpha_i, \beta_i), \qquad (4)$$

where $\alpha_i$ and $\beta_i$ are the parameters of the distribution.

**Overall Framework.** Combining these two ways of leveraging auxiliary data, we now define the likelihood of observed client performance in different rounds, i.e., the round-reputation matrix, as the probability conditioned on the informativeness of the round $r_i$ and the client selection decision $s_j$ (determined by the client utility $\theta_j$):

$$p(\mathbf{R}_{i,j}|r_i, s_j) = r_i^{\mathbb{1}(s_j = \mathbf{R}_{i,j})} + (1 - r_i)^{\mathbb{1}(s_j \neq \mathbf{R}_{i,j})}, \quad (5)$$

where $\mathbb{1}(\cdot)$ is the indicator function. The informativeness

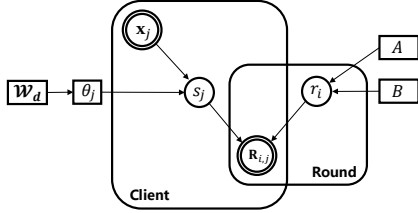

Figure 3: Graphical model of FedTrans. Nodes denoted with circles represent random variables where single circled nodes correspond to hidden variables and doubly circled nodes correspond to observed variables. Nodes denoted with squares refer to the parameters of the model. Edges represent conditional relationships when generating the round-reputation matrix.

of the round is higher if there are more tags $\{\mathbf{R}_{i,j}\}_{j \in \mathcal{J}^i}$ satisfying the actual client utility. The overall framework is depicted in Figure 3. Model updating constitutes parameter learning for $\mathcal{W}_d$ and posterior inference for latent variables $s_j$ and $r_i$.

## 2.2 CLIENT-TRANSPARENT UTILITY ESTIMATION

Parameters of the Bayesian framework are learned by maximizing the likelihood function:

$$p(\mathbf{R}) = \int p(\mathbf{R}, \mathbf{r}, \mathbf{s}|\mathbf{X}; \mathcal{W}_d) d\mathbf{r}, \mathbf{s}, \qquad (6)$$

where $\mathbf{r}$ and $\mathbf{s}$ are the latent true informativeness of all rounds and the selection decision for all clients, respectively; $\mathbf{X}$ is the set of topmost layer $\mathbf{x}_j$ of all local models.

We train the discriminator $f^{\mathcal{W}_d}$ by maximizing the likelihood function in Equation 6. To solve this optimization problem, we transform Equation 6 to the log-likelihood function as

$$\log p(\mathbf{R}) = \underbrace{\int q(\mathbf{r}, \mathbf{s}) \log(\frac{q(\mathbf{r}, \mathbf{s})}{p(\mathbf{r}, \mathbf{s}|\mathbf{R}, \mathbf{X}; \mathcal{W}_d)}) d\mathbf{r}, \mathbf{s}}_{KL(q||p\mathcal{W}_d)} + \underbrace{\int q(\mathbf{r}, \mathbf{s}) \log(\frac{p(\mathbf{R}, \mathbf{r}, \mathbf{s}|\mathbf{X}; \mathcal{W}_d)}{q(\mathbf{r}, \mathbf{s})}) d\mathbf{r}, \mathbf{s}}_{\mathcal{Q}(\mathcal{W}_d, q)} , \quad (7)$$

where $q(\mathbf{r}, \mathbf{s})$ is any probability density function and $KL(\cdot)$ is the Kullback-Leibler (KL) divergence between two distributions.

Maximizing the above function is computationally unfeasible due to the two latent variables in the integral (Tzikas et al., 2008). We solve this problem by Variational Expectation Maximization (variational EM) that iterates between two steps: 1) the E-step where we approximate the distribution of latent variables $p(\mathbf{r}, \mathbf{s}|\mathbf{R}, \mathbf{X}; \mathcal{W}_d)$ with the variational distribution $q(\mathbf{r}, \mathbf{s})$; 2) M-step where we update the estimate for the parameters of the discriminator $\mathcal{W}_d$ by maximizing the evidence lower bound (ELBO) (Blei et al., 2017) of the Equation 6 given the updated latent variables.

**E-Step.** We use the mean-field variational inference approach by assuming that $q(\mathbf{r}, \mathbf{s})$ factorizes over the latent variables:

$$q(\mathbf{r}, \mathbf{s}) = \prod_i q(r_i) \prod_j q(s_j). \quad (8)$$

We further assume each factor function as:

$$q(r_i) = Beta(\alpha_i, \beta_i), \quad (9)$$

$$q(s_j) = Ber(\theta_j), \quad (10)$$

where $\theta_j$, $\alpha_i$, and $\beta_i$ are the variational parameters that can be tuned to minimize the KL-divergence. We use the coordinate ascent to search the optimal parameters, that is, iterating $\theta_j$ ($\alpha_i$, and $\beta_i$) until convergence while keeping $\alpha_i$, and $\beta_i$ ($\theta_j$) fixed.

To update $q(s_j)$, we first keep only the terms that depend on $s_j$ to simplify the $KL(q||p\mathcal{W}_d)$ as

$$q(s_j) \propto p(s_j|\mathbf{x}_j; \mathcal{W}_d) \prod_{i \in \mathcal{I}^j} \exp\{g_{q(r_i)} p(\mathbf{R}_{i,j}|r_i, s_j)\}, \quad (11)$$

where $p(s_j|\mathbf{x}_j; \mathcal{W}_d)$ is the variational distribution of $s_j$ from last iteration, $\mathcal{I}^j$ is the set of rounds containing $j$-th client, and $g_x(\cdot)$ refers to the expectation term $\mathbb{E}_x[log(\cdot)]$ with $x$ being a variational distribution. Based on Equation 11, we can derive the update rule for $q(s_j)$ in Equations 13 and 14.

Following the same idea, we keep only the terms depending on $r_i$ in $KL(q||p\mathcal{W}_d)$ to update the variational distribution $q(r_i)$, and the KL-divergence is simplified as

$$q(r_i) \propto p(r_i) \prod_{j \in \mathcal{J}^i} \exp\{g_{\theta'} p(\mathbf{R}_{i,j}|r_i, s_j)\}, \quad (12)$$

where $p(r_i)$ is the variational distribution of $r_i$ from the last iteration, and $\theta'$ is the estimation of true label distribution in the current iteration. We can derive the update rule for $q(r_i)$ in Equation 15.

The updating rules for $q(\mathbf{r}, \mathbf{s})$ in E-step are given by the following theorems.

**Theorem 2.1 (Incremental Client Utility)** *$q(s_j)$ can be updated based on the output of discriminator $\theta_j$ and the parameters of round informativeness from the rounds containing the $j$-th client $\alpha_i$ and $\beta_i$ ($i \in \mathcal{I}^j$) in the previous iteration. We can derive the update rule*

$$q(s_j = 1) \propto \begin{cases} \theta_j \prod_{i \in \mathcal{I}^j} \exp\{\Psi(\beta_i) - \Psi(\alpha_i + \beta_i)\} & (\mathbf{R}_{i,j} = 0) \\ \theta_j \prod_{i \in \mathcal{I}^j} \exp\{\Psi(\alpha_i) - \Psi(\alpha_i + \beta_i)\} & (\mathbf{R}_{i,j} = 1), \end{cases} \quad (13)$$

$$q(s_j = 0) \propto \begin{cases} (1 - \theta_j) \prod_{i \in \mathcal{I}^j} \exp\{\Psi(\alpha_i) - \Psi(\alpha_i + \beta_i)\} & (\mathbf{R}_{i,j} = 0) \\ (1 - \theta_j) \prod_{i \in \mathcal{I}^j} \exp\{\Psi(\beta_i) - \Psi(\alpha_i + \beta_i)\} & (\mathbf{R}_{i,j} = 1), \end{cases} \quad (14)$$

*where $\Psi(\cdot)$ is the Digamma function.*

---

**Algorithm 1** Variational Utility Inference

---

**Require:** Local updates $\{\mathcal{W}_{i,j}^*\}_{j \in \mathcal{J}^i}$, global model $\bar{\mathcal{W}}_{i-1}^*$, Round-Reputation Matrix **R**, Server auxiliary dataset $\mathcal{D}_a$

1: initialize $\mathcal{W}_d$
2: $\{\mathbf{x}_j\}_{j \in \mathcal{J}^i} \leftarrow$ top-layer of $\{\mathcal{W}_{i,j}^*\}_{j \in \mathcal{J}^i}$
3: $\{\mathbf{x}^+, \mathbf{x}^-\} \leftarrow$ top-layer of weight parameters of synthetic clients initialized by $\bar{\mathcal{W}}_{i-1}^*$ trained on $\mathcal{D}_a^+$ and $\mathcal{D}_a^-$
4: **while** Equation 6 has not converged **do**                  ▷ Variational EM
5:     **while** not converged **do**                          ▷ E-Step
6:         **for** $j \in \mathcal{J}^i$ **do**
7:             update $q(s_j)$ following Equations 13 and 14
8:         **end for**
9:         **for** each $round = 1, 2, \cdots, i$ **do**
10:            update $q(r_i)$ following Equation 15
11:        **end for**
12:    **end while**
13:    **while** not converged **do**                          ▷ M-Step
14:        $\mathcal{W}_d \leftarrow$ training on $\{\mathbf{x}_j\}_{j \in \mathcal{J}^i}$ with label $q(s_j)$ and $\{\mathbf{x}^+, \mathbf{x}^-\}$ with known labels
15:    **end while**
16: **end while**
17: **return** Parameter of the discriminator $\mathcal{W}_d$

---

**Theorem 2.2 (Incremental Round Informativeness)** *The informativeness distribution of $i$-th round $q(r_i)$ can be updated based on parameters $\alpha_i$ and $\beta_i$ from the last E-M iteration and true distribution of client utility in current iteration $\theta'$:*

$$q(r_i) \propto \begin{cases} Beta(\alpha_i + \sum_{j \in \mathcal{J}^i}(1 - \theta'), \beta_i + \sum_{j \in \mathcal{J}^i} \theta') & (\mathbf{R}_{i,j} = 0) \\ Beta(\alpha_i + \sum_{j \in \mathcal{J}^i} \theta', \beta_i + \sum_{j \in \mathcal{J}^i}(1 - \theta')) & (\mathbf{R}_{i,j} = 1). \end{cases} \tag{15}$$

Detailed proof of Theorem 2.1 and Theorem 2.2 can be found in Appendix A.

**M-Step.** Based on the utility of selected clients and rounds quality inferred by the E-step, the target of the M-step is the maximization of the first term in Equation 7:

$$\mathcal{Q}(\mathcal{W}_d, q) = \int q(r_i, s_j) \log p(\mathbf{R}_{i,j}, r_i, s_j | \mathbf{X}; \mathcal{W}_d) dr_i, s_j + const$$

$$= \underbrace{\sum_{s_j} \int q(r_i, s_j) \log p(\mathbf{R}_{i,j} | r_i, s_j) dr_i}_{\mathcal{M}_1} + \underbrace{\sum_{s_j} q(s_j) \log p(s_j | \mathbf{X}; \mathcal{W}_d)}_{\mathcal{M}_2} + const, \tag{16}$$

where $const = \mathbb{E}_{q(r_i, s_j)}[\log \frac{1}{q(r_i, s_j)}]$ is a constant. From the derived equation, only $\mathcal{M}_2$ depends on the parameters $\mathcal{W}_d$ that M-step aims at learning. Obviously, $\mathcal{M}_2$ is the inverse of the cross-entropy between $q(s_j)$ and $p(s_j | \mathbf{X}; \mathcal{W}_d)$, which is used as the loss function for our discriminator. $\mathcal{M}_2$ can, thus, be optimized using standard back-propagation in the case of training the discriminator.

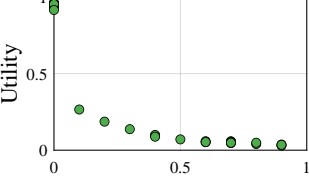

Figure 4: A case study on CIFAR10: utility inference of 30% noisy clients with random local corrupted samples in a proportion from $(0.1, 1]$.

Algorithm 1 describes the entire process of clients' utility inference and Appendix B provides the convergence analysis. In each round, the server receives updates from participating clients as in normal FL settings, as well as simulates the local training on the auxiliary dataset $\mathcal{D}_a$ (row 2-3). It then goes through multiple iterations until convergence (row 4); in each iteration, it incrementally updates the latent variables $s_j$ and $r_i$ in the E-step (row 5-10) and updates $\mathcal{W}_d$ in the M-step (row 11-13). Figure 4 shows an example result of client utility inference where 30% of clients are corrupted by random label flipping

with local noise rates ranging from 0.1 to 1. We observe that FedTrans can effectively estimate the utility of clients, inversely proportional to the actual local noise ratios.

# 3 EXPERIMENTAL RESULTS

## 3.1 EXPERIMENTAL SETUP

**Datasets and Models.** We use two widely-used image datasets: CIFAR10 (Krizhevsky et al., 2009) and Fashion-MNIST (FMNIST) (Xiao et al., 2017). We implement the commonly used neural model for each dataset: LeNet-5 (LeCun et al., 1998) for FMNIST; MobileNetV2 (Sandler et al., 2018) for CIFAR10. We construct the auxiliary dataset by randomly selecting $|\mathcal{D}_a|$ samples from the test set.

**Heterogeneous Data Distribution and Noise Construction.** Inline with recent studies (Kairouz et al., 2019; Li et al., 2022b; Hsu et al., 2019), we create the non-IID data distributions by assigning to the $i$-th client local data that follows the distribution $q_i \in \mathbb{R}^c$, where $c$ is the number of classes and $q_i$ is sampled from a Dirichlet distribution $Dir(\alpha p)$. The concentration parameter $\alpha$ controls the divergence of $q_i$ with respect to a prior distribution $p$ that we set to a uniform distribution. We add noise to local data in both label and feature spaces, as shown in Figure 5.

The first row illustrates three types of label noises constructed in different ways: *Random Flipping* constructs random noise by flipping a label to other classes with equal probabilities (Han et al., 2018; Patrini et al., 2017); *Pair Flipping* constructs structural label noise by assigning labels to the most often confused classes (Rolnick et al., 2017), as determined by a confusion matrix from a centralized training on CIFAR10; *Open-set* constructs label noise by introducing data (features) from another source while keeping the labels unchanged, for instance, a ship image in CIFAR10 is replaced by a bicycle image in CIFAR100 (Tuor et al., 2021; Wang et al., 2018). Feature noises are shown in the bottom row of Figure 5. These include *Gaussian* random noise (with mean 0.2 and variance 1.0), *Corruption* noise where 50% of an image is set to black, and *Resolution* distortion where images are resized to 4×4 and then dilated back to 32×32. We set the above parameters following (Wang et al., 2018).

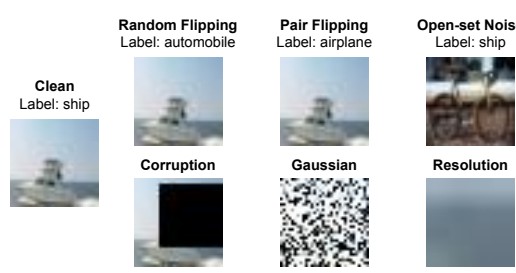

Figure 5: Illustration of data noise in both label and feature space.

**Baselines and Metric** We compare FedTrans with SOTA baselines that: 1) selectively aggregate clients, i.e., FLDebugger (Li et al., 2021), FedCorr (Xu et al., 2022), Oort (Lai et al., 2021) and DivFL (Balakrishnan et al., 2022); 2) adopt robust local objective function, i.e., Robust-FL (Yang et al., 2022b) and RHFL (Fang & Ye, 2022). Besides, we also compare the commonly used vanilla FL framework FedAvg (McMahan et al., 2017) in all settings. For fair comparison, we evaluate FedTrans and baselines on the same test set excluding the auxiliary samples $\mathcal{D}_a$, and further introduce new baselines by fine-tuning FedCorr and DivFL on the auxiliary data. We run all the methods under each setting five times, considering the randomness during model training, and report the average *Top-1 accuracy* within 500 communication rounds. For further evaluation of training efficiency, we also calculate the *wall-clock training time* and *energy-to-accuracy* in the Appendix E and F.

## 3.2 PERFORMANCE EVALUATION

**Resilience to Data Noise.** To evaluate the resilience of FedTrans to data noise, we compare the FedTrans and baselines on the non-iid of Dirichlet distribution with concentration parameter $\alpha = 0.5$ and inject noise to $\epsilon = 30\%$ clients by applying the six label and feature noise creation strategies on the random fraction (from 10% to 100%) of local samples. In each communication round, the server randomly selects 20 participating clients from 100 active ones. We construct the auxiliary dataset $\mathcal{D}_a$ by randomly selecting 200 samples from the test set while keeping a balanced data distribution.

Table 1 reports the results of the compared methods in different noise configurations. We observe that FedTrans consistently outperforms other baselines in all noise configurations on both two tasks. This signifies the ability of FedTrans to deal with noisy clients by providing the server with precise utility inference for client selection. Among the baseline methods, Oort and FLDebugger are

Table 1: Global model accuracy under six types of noise configurations. **Label** noise integrates three label corruptions, **Image** noise integrates three image corruptions, and **Hybrid** noise covers both label and image corruptions, each of which includes two settings: *across-* refers to noisy clients only preserving one of the specified noises, while *intra-* refers to a mix of specified noises in all of the noisy clients. The size of auxiliary data $|\mathcal{D}_a| = 200$. Distribution of the local data is followed by $Dir$ (0.5), and $\epsilon = 30\%$ of active 100 clients are corrupted. We report the average and standard derivation on five trials under each experimental setup.

| | Hybrid (*across-*) | Hybrid (*intra-*) | Label (*across-*) | Label (*intra-*) | Image (*across-*) | Image (*intra-*) |
|---|---|---|---|---|---|---|
| **CIFAR-10, MobileNetV2, *Dir*(0.5)** | | | | | | |
| FedAvg (McMahan et al., 2017) | 69.3% ± 0.6% | 68.3% ± 0.6% | 66.4% ± 0.8% | 66.4% ± 0.3% | 71.0% ± 0.4% | 69.2% ± 2.4% |
| FLDebugger (Li et al., 2021) | 65.0% ± 0.5% | 64.3% ± 0.3% | 66.3% ± 0.2% | 61.2% ± 0.4% | 67.2% ± 0.6% | 66.1% ± 0.5% |
| Oort (Lai et al., 2021) | 63.1% ± 0.5% | 56.2% ± 0.3% | 61.0% ± 1.4% | 56.8% ± 0.8% | 67.6% ± 0.5% | 65.8% ± 0.0% |
| Robust-FL (Yang et al., 2022b) | 73.1% ± 0.3% | 70.6% ± 0.8% | 62.3% ± 0.1% | 73.4% ± 0.4% | 72.5% ± 0.1% | 70.8% ± 0.1% |
| RHFL (Fang & Ye, 2022) | 71.5% ± 0.2% | 70.1% ± 0.1% | 69.4% ± 0.1% | 68.8% ± 0.4% | 72.9% ± 0.5% | 73.0% ± 0.1% |
| DivFL (Balakrishnan et al., 2022) | 73.4% ± 0.2% | 70.1% ± 1.0% | 71.4% ± 0.2% | 70.7% ± 0.3% | 72.7% ± 1.4% | 72.7% ± 0.6% |
| FedCorr (Xu et al., 2022) | 77.0% ± 0.3% | 73.7% ± 0.4% | 72.8% ± 0.1% | **75.7% ± 0.1%** | 73.4% ± 0.3% | 73.7% ± 0.6% |
| Fine-tuned DivFL | 70.6% ± 0.3% | 70.6% ± 0.4% | 69.7% ± 0.3% | 68.7% ± 0.2% | 71.8% ± 0.3% | 70.0% ± 0.4% |
| Fine-tuned FedCorr | 71.1% ± 0.3% | 68.2% ± 0.2% | 70.2% ± 0.3% | 69.2% ± 0.3% | 68.0% ± 0.1% | 67.0% ± 0.2% |
| **FedTrans** | **77.1% ± 0.3%** | **76.9% ± 0.3%** | **76.3% ± 0.2%** | 75.7% ± 0.4% | **77.3% ± 0.1%** | **77.0% ± 0.2%** |
| **FMNIST, LeNet-5, *Dir*(0.5)** | | | | | | |
| FedAvg (McMahan et al., 2017) | 84.9% ± 0.3% | 84.4% ± 0.2% | 83.9% ± 0.2% | 83.6% ± 0.1% | 85.3% ± 0.3% | 85.1% ± 0.3% |
| FLDebugger (Li et al., 2021) | 85.1% ± 0.1% | 85.3% ± 0.1% | 84.9% ± 0.2% | 84.8% ± 0.1% | 84.8% ± 0.2% | 85.1% ± 0.1% |
| Oort (Lai et al., 2021) | 82.6% ± 0.4% | 80.0% ± 0.4% | 80.9% ± 0.9% | 77.0% ± 0.2% | 85.1% ± 0.2% | 85.5% ± 0.0% |
| Robust-FL (Yang et al., 2022b) | 85.0% ± 0.4% | 85.3% ± 0.2% | 84.9% ± 0.1% | 84.8% ± 0.2% | 85.1% ± 0.3% | 85.3% ± 0.1% |
| RHFL (Fang & Ye, 2022) | 84.5% ± 0.0% | 84.6% ± 0.1% | 83.6% ± 0.1% | 84.1% ± 0.1% | 84.5% ± 0.0% | 84.2% ± 0.5% |
| DivFL (Balakrishnan et al., 2022) | 86.1% ± 0.4% | 85.0% ± 0.0% | 85.0% ± 0.0% | 84.8% ± 0.3% | 86.2% ± 0.1% | 86.2% ± 0.0% |
| FedCorr (Xu et al., 2022) | 87.3% ± 0.2% | 87.6% ± 0.1% | 87.3% ± 0.1% | 87.4% ± 0.1% | 86.8% ± 0.8% | 86.1% ± 0.9% |
| Fine-tuned DivFL | 84.3% ± 0.2% | 83.9% ± 0.1% | 83.8% ± 0.1% | 83.5% ± 0.2% | 84.9% ± 0.0% | 84.7% ± 0.1% |
| Fine-tuned FedCorr | 84.9% ± 0.1% | 84.4% ± 0.1% | 83.9% ± 0.0% | 84.6% ± 0.2% | 83.9% ± 0.1% | 84.8% ± 0.1% |
| **FedTrans** | **88.7% ± 0.3%** | **88.0% ± 0.3%** | **88.2% ± 0.2%** | **88.2% ± 0.5%** | **88.6% ± 0.3%** | **88.6% ± 0.5%** |

outperformed by vanilla FedAvG. Their relatively low performance shows that neither the training loss of local models used in Oort nor the difference between the local model weights and global model weights (i.e., $|\mathcal{W}_{i,j}^* - \bar{\mathcal{W}}_i^*|$) used in FLDebugger can approximate utility. In comparison, DivFL shows an improvement by encouraging diverse and representative clients for aggregation (via submodularity-based client selection); FedCorr achieves also a relatively better performance by measuring the local intrinsic data manifold dimensionality (LID) for utility approximation and further clustering data into two subsets (i.e., clean set and noisy set). Most importantly, FedTrans outperforms all comparison methods, including all the above methods and Robust-FL and RHFL whose local objectives are designed to be resilient to the local noise. Furthermore, unlike FedCorr which requires all clients to calculate and report their LID score before FL, FedTrans is flexible to take in new clients during the FL process without extra operations on the client. FedTrans is therefore more favorable in terms of both its transparency and flexibility.

To investigate the ability of FedTrans to exploit auxiliary data, we compare it to the fine-tuned versions of FedCorr and DivFL global models using auxiliary data and report the result also in the table. We conduct multiple trials, considering several epochs of the auxiliary data for fine-tuning the global models in all rounds, and report the best results of the new baselines. We observe that the performance after fine-tuning is notably weaker than before fine-tuning. This highlights the importance of effective strategies to leverage a limited amount of auxiliary data for model improvement. We report further results under different local data distributions in Appendix G.

**Ablation Study.** We conduct it on CIFAR10 with $Dir$(0.5) and hybrid noises across clients, to evaluate each component of FedTrans in client utility estimation. We compare FedTrans's variants with 1) using entries in the $i$-th row of round-reputation matrix $\mathbf{R}$ for client selection, and 2) using outputs of the discriminator, in each round trained on topmost layers of synthetic clients with known labels, for selection. Figure 6 shows the results. We observe FedTrans as a whole, outperforms both variants, verifying the effectiveness of integrating both components for clients' utility estimation.

**Impact of Auxiliary Data Size.** Figure 7 shows the performance of FedTrans given increasing sizes of auxiliary data $|\mathcal{D}_a|$, corresponding to different data partition of $K$ pair(s) of synthetic positive/negative clients (see Section 2.1). We observe no significant accuracy drop until the size of data on the synthetic client becomes too small (i.e., $|\mathcal{D}_a| = 50$ and $K = 3$). The result signifies the cost-efficiency of FedTrans in making use of auxiliary data. We also experiment with a *Misaligned* situation where the noise type in the local data (hybrid noise) is different from that in the manually

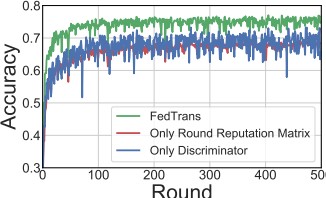

Figure 6: Ablation study on effectiveness of coupling validation & weights discrimination.

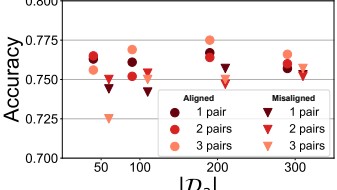

Figure 7: Performance under varied configurations of the auxiliary dataset $\mathcal{D}_a$.

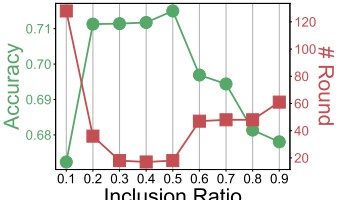

Figure 8: Accuracy and convergence speed when using varied inclusion ratios (0.1 to 0.9).

corrupted $\mathcal{D}^-$ (kept random flipping noise), and observe similar results: no significant performance drop even when the local noise is agnostic.

**Clients Inclusion.** Figure 8 shows the evolution of FedTrans performance under a varying number of clients included in the aggregation ($\epsilon = 60\%$ clients having noise). The global model performance first increases and then decreases with an increasing number of noisy clients involved, which is in line with our previous observations on the effect of noisy clients. This reveals the interesting fact that the global model may benefit from selecting appropriately low-utility updates for aggregation. This phenomenon can be attributed to the benefit of including more training data even with a certain level of noise (e.g., in countering biases of the learning) and can be leveraged for better design of aggregation strategies in the future. In addition to the performance evolution, we also observe a similar trend of the effect of including more noisy clients on the convergence speed.

## 4 RELATED WORK

Learning from noisy data is a critical issue in deep learning (Khetan et al., 2017; Han et al., 2018; Li et al., 2019; 2020), especially for cross-device FL with restricted access to the local data. In the following, we provide a succinct overview of existing literature addressing data noise in FL.

The majority of the work focuses on selecting clean clients. (Lai et al., 2021; Li et al., 2022a) measure the client data utility by the training loss and select clients with large loss. DivFL (Balakrishnan et al., 2022) dynamically clusters clients at each communication round according to the modular score and selects the representative client from each cluster for aggregation. (Cho et al., 2022) provides a theoretical convergence analysis of biased client selection in FL and proposes a selection strategy for trade-off between convergence speed, solution bias, and communication/computation overhead. Our work is close to (Li et al., 2021; Xu et al., 2022) that propose two-step strategies where the server first identifies the corrupted clients, and then guides the clients to rectify the local noisy samples. Note that our FedTrans provides a precise identification of defective clients in a client-transparent way and shows compatibility with the existing sample-level rectification methods.

A few papers propose to design robust local objective functions. Robust-FL (Yang et al., 2022b) introduces into the local loss function a local centroids term to represent and reduce the effect of noisy data on model training. (Fang & Ye, 2022) combines the cross-entropy loss with a reverse term to prevent overfitting to noisy labels. In our experiments, we have shown the better performance of FedTrans over the above methods; note though, that FedTrans can further benefit from the calibration of local objective functions. Finally, our work is related to adversarial attacks (Fung et al., 2020; Tolpegin et al., 2020; Nguyen et al., 2022). The scenarios considered in this work are, however, FL with the presence of natural issues with the local data, instead of those caused by adversarial attacks.

## 5 CONCLUSION

In this paper, we observed that FL clients with noise data slow the learning of global model and reduce its accuracy. To help the server select the most suitable clients for global model aggregation, we proposed a unified Bayesian framework FedTrans to estimate the utility of clients. Our FedTrans is transparent to clients and thus does not require any modifications to clients. Extensive evaluations validate the effectiveness of FedTrans. Our client utility estimation strategy could be leveraged to design better aggregations at the server, to make FL more robust in practical deployments.

ETHICS STATEMENT

We have read the ICLR Code of Ethics and ensured this paper follows it. Our work does not involve dataset releases; all benchmark datasets are publicly available. We anticipate that our work yields a positive societal impact by recognizing corrupted and noisy clients during model aggregation within the Federated Learning framework.

ACKNOWLEDGEMENT

This work is partly funded by the EU's Horizon Europe HarmonicAI project under the HORIZON-MSCA-2022-SE-01 scheme with grant agreement number 101131117. The work is also part of the ICAI GENIUS lab of the research program ROBUST (project number KICH3.LTP.20.006), partly funded by the Dutch Research Council (NWO). We also thank the support provided by Samenwerkende Universitaire RekenFaciliteiten (SURF) with their HPC Cloud infrastructure.

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

# A    Proofs of Theorem 2.1 and Theorem 2.2

## A.1    Preliminaries

FedTrans integrates two client utility estimation strategies: weight-based estimation and performance-based estimation. We formulate a Bayesian framework shown in Figure 3, wherein the parameters are estimated by employing the maximum likelihood estimate (MLE) approach. We first briefly introduce MLE under latent variables settings, and from that derive the parameter estimate rules for our framework. The general MLE problem is formulated as follows:

$$\phi^*_{MLE} = \arg\max_{\phi \in \Phi} p(\mathbf{O}; \phi), \tag{17}$$

where $\phi$ are the parameters of the probabilistic model and $\mathbf{O}$ are a set of observations.

When it is infeasible to directly model the likelihood function $p(\mathbf{O}; \phi)$ (as in our problem), we introduce latent variables $\mathbf{L}$ to connect the observations to unknown parameters. In this way, the likelihood function is transformed into

$$p(\mathbf{O}; \phi) = \int p(\mathbf{O}, \mathbf{L}; \phi) d\mathbf{L}. \tag{18}$$

The expectation maximization (EM) algorithm is employed for MLE with latent variables. The algorithm iteratively updates the elements in the likelihood function. Specifically, the E-step iteratively optimizes the latent variables $\mathbf{L}$, and the M-step iteratively optimizes the parameters $\phi$.

For the problem in this paper, the observations refer to the round reputation matrix $\mathbf{R}$ conditioned on the weight parameters of top-layer $\mathbf{X}$; the latent variables are $\mathbf{r}$ and $\mathbf{s}$; and the parameters are $\{\mathcal{W}_d, A, B\}$, hereafter denoted as $\mathcal{P}$. A detailed expression of the likelihood function is as follows:

$$
\begin{aligned}
p(\mathbf{R}|\mathbf{X}; \mathcal{P}) &= \prod_i \prod_j p(\mathbf{R}_{i,j}|\mathbf{x}_j; \mathcal{P}) \\
&= \prod_i \prod_j \int_{r_i, s_j} p(\mathbf{R}_{i,j}, r_i, s_j|\mathbf{x}_j; \mathcal{P}) dr_i, s_j \\
&= \int p(\mathbf{R}, \mathbf{r}, \mathbf{s}|\mathbf{X}; \mathcal{W}_d) d\mathbf{r}, \mathbf{s}.
\end{aligned} \tag{19}
$$

We first decompose the likelihood function in Equation 6 into two terms as shown in Equation 7 of Section 2.2. We then employ variational EM to optimize variables in the Bayesian framework, wherein the E-step and M-step are designed to iteratively update the variables related to both the two modules. The reason for choosing the variational EM instead of the closed-form EM will also be discussed in the following section.

After applying the logarithm on Equation 19, we get

$$
\begin{aligned}
\log p(\mathbf{R}|\mathbf{X}; \mathcal{P}) &= \log \left( \prod_i \prod_j \int_{r_i, s_j} p(\mathbf{R}_{i,j}, r_i, s_j|\mathbf{x}_j; \mathcal{P}) dr_i, s_j \right) \\
&= \sum_i \sum_j \log \left( \int_{r_i, s_j} p(\mathbf{R}_{i,j}, r_i, s_j|\mathbf{x}_j; \mathcal{P}) dr_i, s_j \right) \\
&= \sum_i \sum_j \log \left( \int_{r_i, s_j} q(r_i, s_j) \frac{p(\mathbf{R}_{i,j}, r_i, s_j|\mathbf{x}_j; \mathcal{P})}{q(r_i, s_j)} dr_i, s_j \right),
\end{aligned} \tag{20}
$$

where the approximation term $q(r_i, s_j)$ can be any probability density function.

According to Jensen's Inequality, we have described the entire process of

$$
\log \left( \int_{r_i, s_j} q(r_i, s_j) \frac{p(\mathbf{R}_{i,j}, r_i, s_j | \mathbf{x}_j; \mathcal{P})}{q(r_i, s_j)} dr_i, s_j \right)
$$
$$
= \log \left( \mathbb{E} \left[ \frac{p(\mathbf{R}_{i,j}, r_i, s_j | \mathbf{x}_j; \mathcal{P})}{q(r_i, s_j)} \right] \right)
$$
$$
\geq \mathbb{E} \left[ \log \left( \frac{p(\mathbf{R}_{i,j}, r_i, s_j | \mathbf{x}_j; \mathcal{P})}{q(r_i, s_j)} \right) \right] \tag{21}
$$
$$
= \int_{r_i, s_j} q(r_i, s_j) \log \left( \frac{p(\mathbf{R}_{i,j}, r_i, s_j | \mathbf{x}_j; \mathcal{P})}{q(r_i, s_j)} \right) dr_i, s_j.
$$

Therefore, we have

$$
\log p(\mathbf{R} | \mathbf{X}; \mathcal{P}) \geq \sum_i \sum_j \int_{r_i, s_j} q(r_i, s_j) \log \left( \frac{p(\mathbf{R}_{i,j}, r_i, s_j | \mathbf{x}_j; \mathcal{P})}{q(r_i, s_j)} \right) dr_i, s_j. \tag{22}
$$

The difference between the two sides of the inequality is

$$
\Delta = \log p(\mathbf{R} | \mathbf{X}; \mathcal{P}) - \sum_i \sum_j \int_{r_i, s_j} q(r_i, s_j) \log \left( \frac{p(\mathbf{R}_{i,j}, r_i, s_j | \mathbf{x}_j; \mathcal{P})}{q(r_i, s_j)} \right) dr_i, s_j
$$
$$
= \log(\prod_i \prod_j p(\mathbf{R}_{i,j} | \mathbf{x}_j; \mathcal{P})) - \sum_i \sum_j \int_{r_i, s_j} q(r_i, s_j) \log \left( \frac{p(\mathbf{R}_{i,j}, r_i, s_j | \mathbf{x}_j; \mathcal{P})}{q(r_i, s_j)} \right) dr_i, s_j
$$
$$
= \sum_i \sum_j \log(p(\mathbf{R}_{i,j} | \mathbf{x}_j; \mathcal{P})) - \sum_i \sum_j \int_{r_i, s_j} q(r_i, s_j) \log \left( \frac{p(\mathbf{R}_{i,j}, r_i, s_j | \mathbf{x}_j; \mathcal{P})}{q(r_i, s_j)} \right) dr_i, s_j
$$
$$
= \sum_i \sum_j \int_{r_i, s_j} \left[ q(r_i, s_j) \log \left( p(\mathbf{R}_{i,j} | \mathbf{x}_j; \mathcal{P}) \right) - q(r_i, s_j) \log \left( \frac{p(\mathbf{R}_{i,j}, r_i, s_j | \mathbf{x}_j; \mathcal{P})}{q(r_i, s_j)} \right) \right] dr_i, s_j
$$
$$
= \sum_i \sum_j \int_{r_i, s_j} q(r_i, s_j) \log \left( \frac{p(\mathbf{R}_{i,j} | \mathbf{x}_j; \mathcal{P}) q(r_i, s_j)}{p(\mathbf{R}_{i,j}, r_i, s_j | \mathbf{x}_j; \mathcal{P})} \right) dr_i, s_j
$$
$$
= \sum_i \sum_j \int_{r_i, s_j} q(r_i, s_j) \log \left( \frac{p(\mathbf{R}_{i,j} | \mathbf{x}_j; \mathcal{P}) q(r_i, s_j)}{p(r_i, s_j | \mathbf{R}_{i,j}, \mathbf{x}_j; \mathcal{P}) p(\mathbf{R}_{i,j} | \mathbf{x}_j; \mathcal{P}))} \right) dr_i, s_j
$$
$$
= \sum_i \sum_j \int_{r_i, s_j} q(r_i, s_j) \log \left( \frac{q(r_i, s_j)}{p(r_i, s_j | \mathbf{R}_{i,j}, \mathbf{x}_j; \mathcal{P}))} \right) dr_i, s_j
$$
$$
= \sum_i \sum_j KL(q || p(r_i, s_j | \mathbf{R}_{i,j}, \mathbf{x}_j; \mathcal{P})). \tag{23}
$$

The gap $\Delta$ refers to the first term in Equation 7, which we need to minimize in the E-step. However, the closed-form EM updates do not work in the case of discrete-continuous variables. We develop a mean field variational inference, following the idea of approximating the posterior distribution of latent variables $p(r_i, s_j | \mathbf{R}_{i,j}, \mathbf{x}_j; \mathcal{P})$ with the variational distribution $q(\mathbf{r}, \mathbf{s}) = \prod_i \prod_j q(r_i, s_j)$.

In the mean-field approach, we assume that

$$
q(\mathbf{r}, \mathbf{s}) = q(\mathbf{r}) q(\mathbf{s}) = \prod_i q(r_i) \prod_j q(s_i), \tag{24}
$$

where $q(r_i) = Beta(\alpha_i, \beta_i)$ (i.e., Equation 9) and $q(s_j) = Ber(\theta_j)$ (i.e., Equation 10).

Next, we will derive the update rule of $q(\mathbf{s})$ and $q(\mathbf{r})$ corresponding to Theorem 2.1 and Theorem 2.2

## A.2   PROOF OF THEOREM 2.1

For $q(\mathbf{s})$, we have

$$
q(\mathbf{s}) \propto \exp\{\mathbb{E}_{q(r_i)}[\log(p(\mathbf{r}, \mathbf{s}, \mathbf{R}, \mathbf{X}; \mathcal{P}))]\}. \tag{25}
$$

For the $j$-th client, there is a set of rounds $\mathcal{I}^j$ involving the $j$-th client. In other words, $\mathbf{R}_{i,j}$ is not blank for $i \in \mathcal{I}^j$. Equation 25 is formulated as

$$
\begin{aligned}
q(s_j) &\propto \exp\{\mathbb{E}_{q(r_i)}[\log(\prod_{i \in \mathcal{I}^j} p(r_i, s_j, \mathbf{R}_{i,j}, \mathbf{x}_j; \mathcal{P}))]\} \\
&\propto \exp\{\mathbb{E}_{q(r_i)}[\sum_{i \in \mathcal{I}^j} \log(p(r_i, s_j, \mathbf{R}_{i,j}, \mathbf{x}_j; \mathcal{P}))]\} \\
&\propto \exp\{\sum_{i \in \mathcal{I}^j} \mathbb{E}_{q(r_i)}[\log(p(r_i, s_j, \mathbf{R}_{i,j}, \mathbf{x}_j; \mathcal{P}))]\} \\
&\propto \prod_{i \in \mathcal{I}^j} \exp\{\mathbb{E}_{q(r_i)}[\log(p(r_i, s_j, \mathbf{R}_{i,j}, \mathbf{x}_j; \mathcal{P}))]\}.
\end{aligned}
\tag{26}
$$

After applying the chain rule on $p(r_i, s_j, \mathbf{R}_{i,j}, \mathbf{x}_j; \mathcal{P})$, we can get

$$
\begin{aligned}
p(r_i, s_j, \mathbf{R}_{i,j}, \mathbf{x}_j; \mathcal{P}) &= p(r_i|\mathbf{x}_j; \mathcal{P}) \times p(s_j|r_i, \mathbf{x}_j; \mathcal{P}) \times p(\mathbf{R}_{i,j}|s_j, r_i, \mathbf{x}_j; \mathcal{P}) \\
&= p(r_i) \times p(s_j|\mathbf{x}_j; \mathcal{P}) \times p(\mathbf{R}_{i,j}|r_i, s_j).
\end{aligned}
\tag{27}
$$

This is because $s_j$ only depends on $\mathbf{x}_j$ and $\mathcal{P}$, and $\mathbf{R}_{i,j}$ does not depend on $\mathbf{x}_j$ and $\mathcal{P}$ given $s_j$ and $r_i$.

Substituting Equation 27 into Equation 26, we have

$$
\begin{aligned}
q(s_j) &\propto \prod_{i \in \mathcal{I}^j} \exp\{\mathbb{E}_{q(r_i)}[\log(p(r_i) \times p(s_j|\mathbf{x}_j; \mathcal{P}) \times p(\mathbf{R}_{i,j}|r_i, s_j))]\} \\
&\propto \prod_{i \in \mathcal{I}^j} \exp\{\mathbb{E}_{q(r_i)}[\log(p(r_i)] + \mathbb{E}_{q(r_i)}[\log(p(s_j|\mathbf{x}_j; \mathcal{P}))] + \mathbb{E}_{q(r_i)}[\log(p(\mathbf{R}_{i,j}|r_i, s_j))]\}.
\end{aligned}
\tag{28}
$$

We remove the irrelevant term related to $q(s_j)$, i.e., $\mathbb{E}_{q(r_i)}[\log(p(r_i)]$, then we get

$$
\begin{aligned}
q(s_j) &\propto \prod_{i \in \mathcal{I}^j} \exp\{\mathbb{E}_{q(r_i)}[\log(p(s_j|\mathbf{x}_j; \mathcal{P}))] + \mathbb{E}_{q(r_i)}[\log(p(\mathbf{R}_{i,j}|r_i, s_j))]\} \\
&\propto \prod_{i \in \mathcal{I}^j} \exp\{\mathbb{E}_{q(r_i)}[\log(p(s_j|\mathbf{x}_j; \mathcal{P}))]\} \times \exp\{\mathbb{E}_{q(r_i)}[\log(p(\mathbf{R}_{i,j}|r_i, s_j))]\}.
\end{aligned}
\tag{29}
$$

Since $\log(p(s_j|\mathbf{x}_j; \mathcal{P}))$ dose not contain the variable $q(r_i)$, we have

$$
\exp\{\mathbb{E}_{q(r_i)}[\log(p(s_j|\mathbf{x}_j; \mathcal{P}))]\} = \exp\{\log(p(s_j|\mathbf{x}_j; \mathcal{P}))\} = p(s_j|\mathbf{x}_j; \mathcal{P}).
\tag{30}
$$

Substituting Equation 30 into Equation 29, we have

$$
\begin{aligned}
q(s_j) &\propto \prod_{i \in \mathcal{I}^j} p(s_j|\mathbf{x}_j; \mathcal{P}) \times \exp\{\mathbb{E}_{q(r_i)}[\log(p(\mathbf{R}_{i,j}|r_i, s_j))]\} \\
&\propto p(s_j|\mathbf{x}_j; \mathcal{P}) \prod_{i \in \mathcal{I}^j} \exp\{\mathbb{E}_{q(r_i)}[\log(p(\mathbf{R}_{i,j}|r_i, s_j))]\}.
\end{aligned}
\tag{31}
$$

Equation 31 is equivalent to Equation 11 of Section 2.2 when we define $\mathbb{E}_{q(r_i)}[\log(\cdot)] = g_{q(r_i)}(\cdot)$.

Based on Equation 31, we now derive the update rule of $q(s_j)$ given the variational parameters $\theta_j$, $\alpha_i$, and $\beta_i$ from last iteration. We first show the proof for $s_j = 1$; the proof for $s_j = 0$ follows similarly.

Equations 2 and 3 of Section 2.1 parameterize the latent variable $s_j$ with the utility $\theta_j$: the selection $s_j$ of $j$-th client follows a Bernoulli distribution parameterized by client utility $\theta_j$ that is the output of a machine learning model $f^{\mathcal{W}_d}(\cdot)$ with input topmost layer $\mathbf{x}_j$ in the $j$-th local model. We have

$$
p(s_j = 1|\mathbf{x}_j; \mathcal{P}) = \theta_j.
\tag{32}
$$

In Equation 5, we connect the round informativeness $r_i$ and client selection $s_j$ under the assumption that rounds with higher informativeness have more entries $R_{i,j}$ satisfying actual client utility, which can be formulated to

$$p(\mathbf{R}_{i,j}|r_i, s_j) = r_i^{(1-|s_j-\mathbf{R}_{i,j}|)} \times (1-r_i)^{|s_j-\mathbf{R}_{i,j}|}. \tag{33}$$

In the case when $s_j = 1$, Equation 33 is equivalent to

$$p(\mathbf{R}_{i,j}|r_i, s_j) = \begin{cases} 1 - r_i & (\mathbf{R}_{i,j} = 0) \\ r_i & (\mathbf{R}_{i,j} = 1). \end{cases} \tag{34}$$

After substituting the probabilities $p(s_j|\mathbf{x}_j; \mathcal{P})$ and $p(\mathbf{R}_{i,j}|r_i, s_j)$ into Equation 31, we get

$$q(s_j = 1) \propto \begin{cases} \theta_j \prod_{i \in \mathcal{I}^j} \exp\{g_{q(r_i)}(1 - r_i)\} & (\mathbf{R}_{i,j} = 0) \\ \theta_j \prod_{i \in \mathcal{I}^j} \exp\{g_{q(r_i)} r_i\} & (\mathbf{R}_{i,j} = 1). \end{cases} \tag{35}$$

By computing the geometric mean of the beta distribution, we can evaluate the expectations $g_x(\cdot)$ as follows:

$$g_{q(r_i)}(1 - r_i) = \Psi(\beta_i) - \Psi(\alpha_i + \beta_i), \tag{36}$$
$$g_{q(r_i)} r_i = \Psi(\alpha_i) - \Psi(\alpha_i + \beta_i). \tag{37}$$

Substituting Equations 36 and 37 into Equation 35, we can obtain the update rule of $q(s_j = 1)$, as given in Equation 13 of Section 2.2 as well as shown below:

$$q(s_j = 1) \propto \begin{cases} \theta_j \prod_{i \in \mathcal{I}^j} \exp\{\Psi(\beta_i) - \Psi(\alpha_i + \beta_i)\} & (\mathbf{R}_{i,j} = 0) \\ \theta_j \prod_{i \in \mathcal{I}^j} \exp\{\Psi(\alpha_i) - \Psi(\alpha_i + \beta_i)\} & (\mathbf{R}_{i,j} = 1). \end{cases}$$

Similarly, for $s_j = 0$, we have

$$p(s_j = 0|\mathbf{x}_j; \mathcal{P}) = 1 - \theta_j, \tag{38}$$

and

$$p(\mathbf{R}_{i,j}|r_i, s_j) = \begin{cases} r_i & (\mathbf{R}_{i,j} = 0) \\ 1 - r_i & (\mathbf{R}_{i,j} = 1). \end{cases} \tag{39}$$

Equation 31 is then equivalent to

$$q(s_j = 0) \propto \begin{cases} (1 - \theta_j) \prod_{i \in \mathcal{I}^j} \exp\{g_{q(r_i)} r_i\} & (\mathbf{R}_{i,j} = 0) \\ (1 - \theta_j) \prod_{i \in \mathcal{I}^j} \exp\{g_{q(r_i)}(1 - r_i)\} & (\mathbf{R}_{i,j} = 1). \end{cases} \tag{40}$$

Again, substituting Equation 36 and 37 into Equation 40, we can obtain the update rule of $q(s_j = 0)$ as Equation 14 of Section 2.2:

$$q(s_j = 0) \propto \begin{cases} (1 - \theta_j) \prod_{i \in \mathcal{I}^j} \exp\{\Psi(\alpha_i) - \Psi(\alpha_i + \beta_i)\} & (\mathbf{R}_{i,j} = 0) \\ (1 - \theta_j) \prod_{i \in \mathcal{I}^j} \exp\{\Psi(\beta_i) - \Psi(\alpha_i + \beta_i)\} & (\mathbf{R}_{i,j} = 1). \end{cases}$$

We now conclude the proof of Theorem 2.1.

### A.3 PROOF OF THEOREM 2.2

For $q(\mathbf{r})$, we have

$$q(\mathbf{r}) \propto \exp\{\mathbb{E}_{q(s_j)}[\log(p(\mathbf{r}, \mathbf{s}, \mathbf{R}, \mathbf{X}; \mathcal{P}))]\}. \tag{41}$$

In the $i$-th round, there is a set of participating clients $\mathcal{J}^i$ corresponding to entries in the $i$-th row of matrix $\mathbf{R}$. Equation 41 is formulated as

$$q(r_i) \propto \exp\{\mathbb{E}_{q(s_j)}[\log(\prod_{j \in \mathcal{J}^i} p(r_i, s_j, \mathbf{R}_{i,j}, \mathbf{x}_j; \mathcal{P}))]\}. \tag{42}$$

Following the transformation of $q(s_j)$, we can also get a simplified version of Equation 42, similar to Equation 31:

$$q(r_i) \propto p(r_i) \prod_{j \in \mathcal{J}^i} \exp\{\mathbb{E}_{q(s_j)}[\log(p(\mathbf{R}_{i,j}|r_i, s_j))]\} \tag{43}$$

Since $q(s_j) = Ber(\theta_j)$, we have

$$
\begin{aligned}
\mathbb{E}_{q(s_j)}[\log(p(\mathbf{R}_{i,j}|r_i, s_j))] &= \sum_{s_j} q(s_j)\log(p(\mathbf{R}_{i,j}|r_i, s_j)) \\
&= q(s_j = 0) \times \log(p(\mathbf{R}_{i,j}|r_i, s_j)) + q(s_j = 1) \times \log(p(\mathbf{R}_{i,j}|r_i, s_j)) \\
&= (1 - \theta_j) \times \log(p(\mathbf{R}_{i,j}|r_i, s_j)) + \theta_j \times \log(p(\mathbf{R}_{i,j}|r_i, s_j)) \\
&\stackrel{def}{=} \mathbb{E}_{\theta_j}[\log(p(\mathbf{R}_{i,j}|r_i, s_j))].
\end{aligned}
\tag{44}
$$

Therefore, Equation 43 is finally transformed into

$$q(r_i) \propto p(r_i) \prod_{j \in \mathcal{J}^i} \exp\{\mathbb{E}_{\theta_j}[\log(p(\mathbf{R}_{i,j}|r_i, s_j))]\}. \tag{45}$$

Equation 45 is equivalent to Equation 12 of Section 2.2 when we define $\mathbb{E}_{\theta_j}[log(\cdot)] = g_{\theta'}(\cdot)$.

Based on Equation 45, we now derive the update rule of $q(r_i)$ given the variational parameters $\theta_j$, $\alpha_i$, and $\beta_i$ from last iteration.

We replace the probability $p(r_i)$ in Equation 45 by the Beta distribution with parameters $\alpha_i$ and $\beta_i$ from the previous iteration:

$$q(r_i) \propto Beta(\alpha_i, \beta_i) \prod_{j \in \mathcal{J}^i} \exp\{\mathbb{E}_{\theta_j}[\log(p(\mathbf{R}_{i,j}|r_i, s_j))]\}. \tag{46}$$

According to Equation 33, the $\exp\{\mathbb{E}_{\theta_j}[\log(p(\mathbf{R}_{i,j}|r_i, s_j))]\}$ in Equation 46 can be reformulated to

$$
\begin{aligned}
&\exp\{\mathbb{E}_{\theta_j}[\log(p(\mathbf{R}_{i,j}|r_i, s_j))]\} \\
&= \exp\{\mathbb{E}_{\theta_j}[\log(r_i^{(1-|s_j-\mathbf{R}_{i,j}|)} \times (1-r_i)^{|s_j-\mathbf{R}_{i,j}|})]\} \\
&= \exp\{(1-\theta_j) \times \log(r_i^{(1-|\mathbf{R}_{i,j}|)} \times (1-r_i)^{|\mathbf{R}_{i,j}|}) + \theta_j \times \log(r_i^{(1-|1-\mathbf{R}_{i,j}|)} \times (1-r_i)^{|1-\mathbf{R}_{i,j}|})\} \\
&= \exp\{(1-\theta_j) \times \log(r_i^{(1-\mathbf{R}_{i,j})} \times (1-r_i)^{\mathbf{R}_{i,j}}) + \theta_j \times \log(r_i^{(\mathbf{R}_{i,j})} \times (1-r_i)^{(1-\mathbf{R}_{i,j})})\} \\
&= \exp\{(1-\theta_j) \times \log(r_i^{(1-\mathbf{R}_{i,j})} \times (1-r_i)^{\mathbf{R}_{i,j}})\} \times \exp\{\theta_j \times \log(r_i^{(\mathbf{R}_{i,j})} \times (1-r_i)^{(1-\mathbf{R}_{i,j})})\} \\
&= (r_i^{(1-\mathbf{R}_{i,j})} \times (1-r_i)^{\mathbf{R}_{i,j}})^{(1-\theta_j)} \times (r_i^{\mathbf{R}_{i,j}} \times (1-r_i)^{(1-\mathbf{R}_{i,j})})^{\theta_j} \\
&= r_i^{(1+2\mathbf{R}_{i,j}\theta_j-\theta_j-\mathbf{R}_{i,j})}(1-r_i)^{(\mathbf{R}_{i,j}-2\mathbf{R}_{i,j}\theta_j+\theta_j)}.
\end{aligned}
\tag{47}
$$

Therefore, the expectation term in Equation 46 can be evaluated as follows:

$$\exp\{\mathbb{E}_{\theta_j}[\log(p(\mathbf{R}_{i,j}|r_i, s_j))]\} = \begin{cases} r_i^{(1-\theta_j)}(1-r_i)^{\theta_j} & (\mathbf{R}_{i,j} = 0) \\ r_i^{\theta_j}(1-r_i)^{(1-\theta_j)} & (\mathbf{R}_{i,j} = 1). \end{cases} \tag{48}$$

In the case when $\mathbf{R}_{i,j} = 1$, Equation 46 is equivalent to :

$$q(r_i) \propto Beta(\alpha_i, \beta_i) \prod_{j \in \mathcal{J}^i} r_i^{\theta_j}(1-r_i)^{(1-\theta_j)}. \tag{49}$$

The probability density function of $r_i$'s distribution is given by:

$$Beta(\alpha_i, \beta_i) \propto r_i^{(\alpha_i-1)}(1-r_i)^{(\beta_i-1)}. \tag{50}$$

Substituting Equation 50 into Equation 49, we get

$$
\begin{aligned}
q(r_i) &\propto r_i^{(\alpha_i-1)}(1-r_i)^{(\beta_i-1)} \prod_{j\in\mathcal{J}^i} r_i^{\theta_j}(1-r_i)^{(1-\theta_j)} \\
&\propto \prod_{j\in\mathcal{J}^i} r_i^{(\alpha_i-1)}(1-r_i)^{(\beta_i-1)} r_i^{\theta_j}(1-r_i)^{(1-\theta_j)} \\
&\propto \prod_{j\in\mathcal{J}^i} r_i^{(\alpha_i+\theta_j-1)}(1-r_i)^{(\beta_i+(1-\theta_j)-1)} \\
&\propto r_i^{(\alpha_i+\sum_{j\in\mathcal{J}^i}\theta_j-1)}(1-r_i)^{(\beta_i+\sum_{j\in\mathcal{J}^i}(1-\theta_j)-1)} \\
&\propto Beta\left(\alpha_i+\sum_{j\in\mathcal{J}^i}\theta_j,\ \beta_i+\sum_{j\in\mathcal{J}^i}(1-\theta_j)\right).
\end{aligned}
\tag{51}
$$

Similarly, for $\mathbf{R}_{i,j}=0$, we have

$$
q(r_i) \propto Beta(\alpha_i,\beta_i) \prod_{j\in\mathcal{J}^i} r_i^{(1-\theta_j)}(1-r_i)^{\theta_j}.
\tag{52}
$$

Again, substituting Equation 50 into Equation 52, we complete the proof of Theorem 2.2 as follows:

$$
\begin{aligned}
q(r_i) &\propto r_i^{(\alpha_i-1)}(1-r_i)^{(\beta_i-1)} \prod_{j\in\mathcal{J}^i} r_i^{(1-\theta_j)}(1-r_i)^{\theta_j} \\
&\propto \prod_{j\in\mathcal{J}^i} r_i^{(\alpha_i-1)}(1-r_i)^{(\beta_i-1)} r_i^{(1-\theta_j)}(1-r_i)^{\theta_j} \\
&\propto \prod_{j\in\mathcal{J}^i} r_i^{(\alpha_i+(1-\theta_j)-1)}(1-r_i)^{(\beta_i+\theta_j-1)} \\
&\propto r_i^{(\alpha_i+\sum_{j\in\mathcal{J}^i}(1-\theta_j)-1)}(1-r_i)^{(\beta_i+\sum_{j\in\mathcal{J}^i}\theta_j-1)} \\
&\propto Beta\left(\alpha_i+\sum_{j\in\mathcal{J}^i}(1-\theta_j),\ \beta_i+\sum_{j\in\mathcal{J}^i}\theta_j\right).
\end{aligned}
\tag{53}
$$

## B  CONVERGENCE ANALYSIS

### B.1  PRELIMINARIES

Before the convergence analysis on FedTrans, we first provide a detailed formulation of the local training and global aggregation in FedTrans with selective client participation.

In the $i$-th communication round, the cloud server randomly selects a set of participating clients $\mathcal{J}^i$. Each client $j\in\mathcal{J}^i$ performs the stochastic gradient descent (SGD) on the local data instances:

$$
\mathcal{L}(f_j^{\mathcal{W}_{\tau,j}},\xi_\tau^j) = \frac{1}{|\xi_\tau^j|}\sum_{x\in\xi_\tau^j} l(f_j^{\mathcal{W}_{\tau,j}};x),
\tag{54}
$$

$$
\hat{\mathcal{W}}_{\tau+1,j} = \mathcal{W}_{\tau,j} - \eta_\tau \nabla\mathcal{L}(f_j^{\mathcal{W}_{\tau,j}},\xi_\tau^j),
\tag{55}
$$

where $\tau$ indexes the local SGD step on the objective $\mathcal{L}(\cdot)$ regarding the loss function $l(\cdot)$ (e.g., cross-entropy loss). $\xi_\tau^j$ is a batch of samples randomly selected from local data $\mathcal{D}_j$ at step $\tau$. Client updates local model to $f_j^{\hat{\mathcal{W}}_{\tau+1,j}}$ after one-step gradient descent with the learning rate $\eta_\tau$. Suppose that client reports the model updates every $E\in\mathbb{Z}^+$ steps, and we have the update rule

$$
\mathcal{W}_{\tau+1,j} = \begin{cases} \hat{\mathcal{W}}_{\tau+1,j}, & \tau+1\neq iE \\ \sum_{j\in\mathcal{J}^i} p_j\hat{\mathcal{W}}_{\tau+1,j}, & \tau+1=iE. \end{cases}
\tag{56}
$$

We define the local updates of client $j$ offloading to the cloud server in round $i$ as

$$\mathcal{W}_{i,j}^* \triangleq \hat{\mathcal{W}}_{iE,j}. \tag{57}$$

Under the guidance of client utility estimated by FedTrans, the server aggregates a subset of updates from clients $\hat{\mathcal{J}}^i \subseteq \mathcal{J}^i$ for the global model

$$\bar{\mathcal{W}}_i^* \triangleq \sum_{j \in \hat{\mathcal{J}}^i} \hat{p}_j \mathcal{W}_{i,j}^* = \sum_{j \in \hat{\mathcal{J}}^i} \hat{p}_j \hat{\mathcal{W}}_{iE,j}. \tag{58}$$

In the following section, we simplify the objective function as $\mathcal{L}_j(\mathcal{W}, \xi) = \mathcal{L}(f_j^{\mathcal{W}}, \xi)$ and gradients as $g_j(\mathcal{W}, \xi) = \nabla\mathcal{L}_j(\mathcal{W}, \xi)$ for a more concise expression. Specifically, $\mathcal{L}_j(\mathcal{W}) = \mathcal{L}(f_j^{\mathcal{W}}, \mathcal{D}_j)$ and $g_j(\mathcal{W}) = \nabla\mathcal{L}_j(\mathcal{W})$.

## B.2 CONVERGENCE GUARANTEE

A rigorous convergence analysis of FedTrans is non-trivial since the tendentious selection of clients with high utility compared with random client selection (e.g., FedAvg). Borrowing from the theoretical analysis in (Cho et al., 2022) that considers the effect of biased client participation on convergence, we can provide the convergence guarantee of FedTrans under assumptions

**Assumption 1** $\mathcal{L}_1, \cdots, \mathcal{L}_{|\mathcal{J}|}$ *are all $L$-smooth, i.e., for all $\mathcal{W}$ and $\mathcal{W}'$,*

$$\mathcal{L}_j(\mathcal{W}) \leq \mathcal{L}_j(\mathcal{W}') + \langle \nabla\mathcal{L}_j(\mathcal{W}'), \mathcal{W} - \mathcal{W}' \rangle + \frac{L}{2}\|\mathcal{W} - \mathcal{W}'\|_2^2. \tag{59}$$

**Assumption 2** $\mathcal{L}_{\mathcal{D}_1}, \cdots, \mathcal{L}_{\mathcal{D}_{|\mathcal{J}|}}$ *are all $\mu$-strongly convex, i.e., for all $\mathcal{W}$ and $\mathcal{W}'$,*

$$\mathcal{L}_j(\mathcal{W}) \geq \mathcal{L}_j(\mathcal{W}') + \langle \nabla\mathcal{L}_j(\mathcal{W}'), \mathcal{W} - \mathcal{W}' \rangle + \frac{\mu}{2}\|\mathcal{W} - \mathcal{W}'\|_2^2. \tag{60}$$

**Assumption 3** *For mini-batch $\xi_\tau^j$ uniformly sampled at random from local data of $j$-th client $\mathcal{D}_j$, the resulting stochastic gradient is unbiased, that is, $\mathbb{E}[g_j(\mathcal{W}_{\tau,j}, \xi_\tau^j)] = g_j(\mathcal{W}_{\tau,j})$. Also, the variance of the stochastic gradient is bounded, i.e., for all $j = 1, \cdots, |\mathcal{J}|$ and $\forall\tau$,*

$$\mathbb{E}[\|g_j(\mathcal{W}_{\tau,j}, \xi_\tau^j) - g_j(\mathcal{W}_{\tau,j})\|^2] \leq \sigma^2. \tag{61}$$

**Assumption 4** *The stochastic gradient's expected squared norm is uniformly bounded, i.e., for all $j = 1, \cdots, |\mathcal{J}|$ and $\forall\tau$,*

$$\mathbb{E}[\|g_j(\mathcal{W}_{\tau,j}, \xi_\tau^j)\|^2] \leq G^2. \tag{62}$$

In line with (Cho et al., 2022), we then define two metrics: local-global objective gap and selection skew. The local-global objective gap is formulated as

$$\Gamma \triangleq \sum_j^{|\mathcal{J}|} p_j(\mathcal{L}_j(\bar{\mathcal{W}}^*) - \mathcal{L}_j(\mathcal{W}_j^*)), \tag{63}$$

where $p_j$ refers to the fraction of data at $j$-th client to the overall data volume. The $\bar{\mathcal{W}}^*$ and $\mathcal{W}_j^*$ are the optimal weight parameters of the global objective $\mathcal{L}(\cdot)$ and local objective $\mathcal{L}_j(\cdot)$ respectively. To be more specific,

$$\mathcal{W}_j^* = \arg\min_{\mathcal{W}} \mathcal{L}_j(\mathcal{W}), \tag{64}$$

$$\bar{\mathcal{W}}^* = \arg\min_{\mathcal{W}} \mathcal{L}(\mathcal{W}) = \arg\min_{\mathcal{W}} \sum_j^{|\mathcal{J}|} p_j\mathcal{L}_j(\mathcal{W}). \tag{65}$$

Further, we denote $\mathcal{L}_j^* = \mathcal{L}_j(\mathcal{W}_j^*)$ and $\mathcal{L}^* = \mathcal{L}(\bar{\mathcal{W}}^*)$.

The selection skew is formulated as

$$\rho(\mathcal{S}(\pi, \mathcal{W}), \mathcal{W}') = \frac{\mathbb{E}_{\mathcal{S}(\pi, \mathcal{W})}[\frac{1}{m} \sum_{j \in \mathcal{S}(\pi, \mathcal{W})}(\mathcal{L}_j(\mathcal{W}') - \mathcal{L}_j^*)]}{\mathcal{L}(\mathcal{W}') - \sum_j^{|\mathcal{J}|} p_j \mathcal{L}_j^*} \tag{66}$$

where $\mathcal{S}(\pi, \mathcal{W})$ is a set of selected $m$ clients given selection strategy $\pi$ according to weight parameters $\mathcal{W}$, and $\mathcal{W}'$ is the observing point where we evaluate the local objective $\mathcal{L}_j(\mathcal{W}')$ and global objective $\mathcal{L}(\mathcal{W}')$.

We then define two related metrics

$$\bar{\rho} \triangleq \min_{\mathcal{W}, \mathcal{W}'} \rho(\mathcal{S}(\pi, \mathcal{W}), \mathcal{W}'), \tag{67}$$

$$\tilde{\rho} \triangleq \max_{\mathcal{W}} \rho(\mathcal{S}(\pi, \mathcal{W}), \mathcal{W}^*). \tag{68}$$

Under the above-mentioned three assumptions, for learning rate $\eta_t = \frac{1}{\mu(t+\gamma)}$ and $\gamma = \frac{4L}{\mu}$, we have the convergence with any selection strategy $\pi$ after $T$ local steps as

$$\mathbb{E}[\mathcal{L}(\bar{\mathcal{W}}_T^*)] - \mathcal{L}^* \leqslant \frac{1}{T+\Gamma}\Big[\frac{4L(32\tau^2 G^2 + \sigma^2/m)}{3\mu^2\bar{\rho}} + \frac{8L^2\Gamma}{\mu^2} + \frac{L\gamma\|\bar{\mathcal{W}}_0^* - \bar{\mathcal{W}}^*\|^2}{2}\Big] + \frac{8L\Gamma}{3\mu}(\frac{\tilde{\rho}}{\bar{\rho}} - 1) \tag{69}$$

where $T$ is the local SGD interactions, and $G$ is the upper bound of the stochastic gradient's expected squared norm in assumption (3).

Similarly, for a fixed learning rate $\eta \leqslant \min\{\frac{1}{2\mu B}, \frac{1}{4L}\}$ where is $B = 1 + \frac{3\bar{\rho}}{8}$, we have the convergence with any selection strategy $\pi$ as

$$\mathbb{E}[\mathcal{L}(\bar{\mathcal{W}}_T^*)] - \mathcal{L}^* \leqslant \frac{4L\eta(32\tau^2 G^2 + \frac{\sigma^2}{m}) + 6\bar{\rho}L\Gamma}{\mu(8 + 3\bar{\rho})} + \frac{8L\Gamma(\tilde{\rho} - \bar{\rho})}{\mu(8 + 3\bar{\rho})}$$
$$+ \frac{L}{\mu}\left(1 - \eta\mu(1 + \frac{3\bar{\rho}}{8})\right)^T \left(\mathcal{L}(\bar{\mathcal{W}}_0^*) - \mathcal{L}^* - \frac{4\left(\eta(32\tau^2 G^2 + \frac{\sigma^2}{m} + 6\bar{\rho}L\Gamma) + 2\Gamma(\tilde{\rho} - \bar{\rho})\right)}{8 + 3\bar{\rho}}\right). \tag{70}$$

For a small $\eta$, both fixed-learning rate case and decaying-learning rate case have the same upper bound by $\frac{8L\Gamma}{3\mu}(\frac{\tilde{\rho}}{\bar{\rho}} - 1)$.

In FedTrans, the selection strategy $\pi_{util}$ chooses clients $\hat{\mathcal{J}}^i \subseteq \mathcal{J}^i$ from the participating clients based on the estimated utilities. Therefore, in Equation 66, $\mathcal{S}(\pi, \mathcal{W}) = \hat{\mathcal{J}}^i$ and $m = |\hat{\mathcal{J}}^i|, \forall i \in \mathcal{I}$. According to the analysis in (Cho et al., 2022), a larger selection skew $\rho$ results in faster convergence. We provide the lower bound of convergence rate at $\mathcal{O}(\frac{1}{T\bar{\rho}})$ where $T$ denotes the accumulated local SGD steps, since $\bar{\rho}$ calculates the minimum of the selection skew, as shown in Equation 67. In practice, the varying weights parameters of the global model $\bar{\mathcal{W}}_i^*$ and the local model $\mathcal{W}_{i,j}^*$ cause the selection skew $\rho(\mathcal{S}(\pi, \bar{\mathcal{W}}_i^*), \mathcal{W}_{i,j}^*)$ to change with the FL proceeding while maintaining convergence rate of at least of $\bar{\rho}$.

## C MORE EXPERIMENTAL DETAILS

**Implementation Details.** The parameters of variational utility inference and those for discriminator $f^{\mathcal{W}_d}$ training are empirically set. We adopt $f^{\mathcal{W}_d}$ with Multi-Layer Perception (MLP) having 2 hidden layers of 128 and 64 dimensions respectively. In discriminator training, we select the learning rate as $1e - 3$, and we set the priors $A$ and $B$ by sampling from a uniform distribution $\sim [0, 10]$ and update them in E-step according to Theorem 2.1 and Theorem 2.2.

**Comparison Methods.** We implement all the comparison methods in Python and the neural networks with PyTorch, running on an NVIDIA 2080Ti GPU. In local training, local epochs are set to 5 and the learning rate is $1e - 2$. We use SGD with momentum factor = 0.9 as the local optimizer.

To evaluate the resilience of FedTrans to data noise, we compare it with SOTA baselines. RHFL (Fang & Ye, 2022) considers symmetrically using cross-entropy loss and reverse cross-entropy loss to ameliorate the negative effect of internal local model noise. Robust-FL (Yang et al., 2022b) copes with the noisy federated setting by interchanging additional information called class-wise centroids. FLDebugger (Li et al., 2021) utilizes the 2-norm distance between local weight parameters and global weight parameters to distinguish noisy clients. FedCorr (Xu et al., 2022) calculates the average Local Intrinsic Dimension (LID) of local prediction vectors for each client. The server applies a Gaussian Mixture Model on received LID scores to partition involved clients into two subsets: noisy clients and clean clients. As a client selection method, Oort (Lai et al., 2021) achieves enhanced time-to-accuracy by constructing both statistical utility and system utility for clients. DivFL (Balakrishnan et al., 2022) selects a subset of clients whose weight updates closely mimic the information gained from aggregating updates across all clients, to improve learning efficiency. Note that FLDebugger and FedCorr are sample-wise noisy correction methods where we partially compare two baselines with regard to the identification of noisy clients. We implement the component about data utility in Oort for comparison.

**Observation details.** Figure 1 illustrates the negative impact of noisy clients under two different local data distributions, emphasizing more severe degradation in global model performance when local data is non-IID across clients. For the completeness of experiments, we also explore the performance of the global model varying with learning rates other than $10^{-2}$ in Figure 1. Compared with results in Figure 9, under both IID and non-IID settings, the global model achieves the best performance when the learning rate is $10^{-2}$ as used in Figure 1. Furthermore, we observe a more pronounced degradation of the model performance introduced by data noise in non-IID data compared to IID data across all learning rates ranging from $10^{-1}$ to $10^{-4}$, which necessitates inferring client utility in more realistic non-IID scenarios.

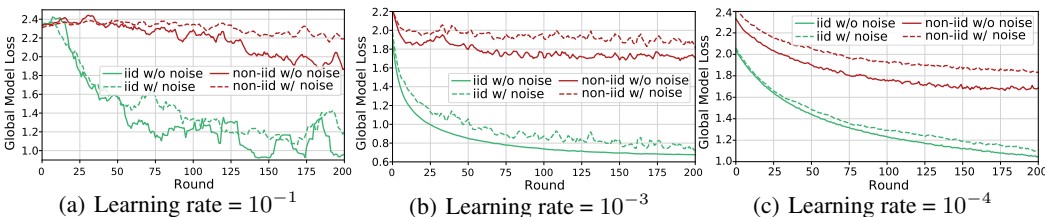

(a) Learning rate = $10^{-1}$      (b) Learning rate = $10^{-3}$      (c) Learning rate = $10^{-4}$

Figure 9: Global model performance of FL with noisy clients (in Hybrid *(across-)* local noise). We consider two data distributions and three different learning rates.

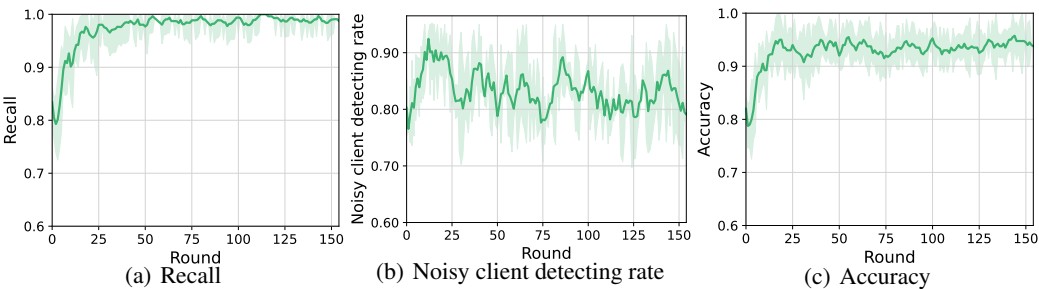

(a) Recall      (b) Noisy client detecting rate      (c) Accuracy

Figure 10: The client selection performance in the presence of $\epsilon = 30\%$ noisy clients.

**Client selection performance.** In the experiments, we adopt a threshold-based selection strategy, excluding the updates of the clients with the utility $\theta$ below 0.5 from the global model aggregation. As shown in Figure 10, we evaluate the client selection performance in the setting where $\epsilon = 30\%$ clients corrupted by Hybrid *(across-)* noise. We observe the high recall performance, stabilizing at

approximately 99% after 50 communication rounds. It indicates FedTrans's effective recognition of clean (positive) clients on the server side. The success rate of detecting noisy clients (true negative rate) reaches an average of 85% as the rounds progress. Furthermore, the overall accuracy of client selection according to the utilities estimated by FedTrans is around 95%, given that the clean clients are the majority participants (at approximately 70%) in each communication round.

**Correlation between utility and noise rate.** As inferring client utility is the main focus of this paper, we investigate the performance of utility estimation by FedTrans across communication rounds. In contrast to evaluating the success rate of detecting noisy clients, our assessment measures the strength of the monotonic relationship between estimated utility and actual local noise. Disregarding the specific selection strategy (e.g., the threshold-based method) according to client utilities, we employ Spearman's rank-order correlation coefficient (PROCC) to quantify the statistical dependence between two variables (i.e., $\theta_j$ and local noise rate). As shown in Figure 11, the utility exhibits a negative relationship with local noise, and this relationship becomes more significant along with FL progress. This observation indicates that utility estimation becomes more accurate and aligned with the client noise degree,

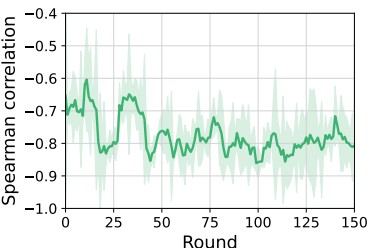

Figure 11: The correlation between estimated client utility and its actual local noise rate varies with the communication rounds.

primarily attributed to the incremental updates on the discriminator $f^{\mathcal{W}_d}$ across communication rounds.

## D    TRANSPARENT CLIENT UTILITY ESTIMATION IN FEDERATED LEARNING

Here, we provide the pseudo-code of the overall FL process with FedTrans as a general algorithmic framework (see Algorithm 2). In each round, local models in participating clients are first updated as in standard FL settings (**rows 3-6**). The server then updates the round-reputation matrix and infers client utility (**row 8-11**) by calling the variational utility inference algorithm (see Algorithm 1). After that, it can perform an arbitrary client selection strategy guided by estimated client utility (**row 12-14**). It finally obtains the global model by aggregating the weight parameters from selected clients (**row 15**). FedTrans, as a module that does not require any additional information from the client, can be coupled to any existing aggregation and local training schemes.

---

**Algorithm 2** Federated Learning with FedTrans

---

1: **Require:** A set of clients with self-contained data: $\mathcal{C}$; Server auxiliary dataset: $\mathcal{D}_a$; Client selection rate: $\gamma$.
2: **for** each round $i = 1, 2, \cdots, N$ **do**
3:      $\mathcal{J}^i \leftarrow$ randomly select $max(|\mathcal{C}| \times \gamma, 1)$ clients from $\mathcal{C}$
4:      **for** $j \in \mathcal{J}^i$ **in parallel do**                                                           ▷ Local Training
5:          $\mathcal{W}_{i,j}^* \leftarrow ClientUpdate\,(C_j, \bar{\mathcal{W}}_{i-1}^*)$
6:      **end for**
7:      **Server Executes FedTrans:**
8:      $\mathbf{R}_i \leftarrow MatrixUpdate\,(\{\mathcal{W}_{i,j}^*\}_{j \in \mathcal{J}^i}, \mathbf{R}_{i-1}, \mathcal{D}_a)$
9:      $\mathcal{W}_d \leftarrow VariationalInference\,(\{\mathcal{W}_{i,j}^*\}_{j \in \mathcal{J}^i}, \bar{\mathcal{W}}_{i-1}^*, \mathbf{R}_i)$          ▷ Utility Estimation
10:     $\{\mathbf{x}_j\}_{j \in \mathcal{J}^i} \leftarrow$ top-layer of $\{\mathcal{W}_{i,j}^*\}_{j \in \mathcal{J}^i}$
11:     $\{\theta_j\}_{j \in \mathcal{J}^i} \leftarrow f^{\mathcal{W}_d}(\{\mathbf{x}_j\}_{j \in \mathcal{J}^i})$
12:     **for** $j \in \mathcal{J}^i$ **do**
13:         $s_j \leftarrow$ client selection guided by $\theta_j$                                          ▷ Client Selection
14:     **end for**
15:     $\bar{\mathcal{W}}_i^* \leftarrow Aggregation\,(\{\mathcal{W}_{i,j}^*\}_{j \in \mathcal{J}^i}, \{s_j\}_{j \in \mathcal{J}^i})$          ▷ Server Aggregation
16: **end for**

---

# E  TIME CONSUMPTION

Although the FedTrans is running on the server with relatively rich computing resources, this inevitably incurs extra training time overheads. We present a detailed time consumption analysis under noisy clients (i.e., Hybrid (cross-)) on the CIFAR10 dataset with distribution Dir(0.5), as shown in Table 2. Note that FedCorr Xu et al. (2022) requires involving all clients to estimate a set of noisy clients before Federated Learning (FL) training starts. Therefore, we do not investigate the per-round time consumption of FedCorr in this context.

Table 2: Time consumption per round or when achieving the target accuracy for each method.

| | CIFAR-10, MobileNetV2, *Dir*(0.5) | | | | | | |
| --- | --- | --- | --- | --- | --- | --- | --- |
| | **FedAvg** | **FLDebugger** | **Oort** | **Robust-FL** | **RHFL** | **DivFL** | **FedTrans** |
| Per round (seconds) | $113 \pm 3$ | $115 \pm 2$ | $114 \pm 3$ | $129 \pm 8$ | $115 \pm 3$ | $114 \pm 3$ | $173 \pm 5$ |
| Target accuracy (minutes) | $118.7 \pm 3.2$ | $280.8 \pm 4.9$ | $385.8 \pm 10.2$ | $60.9 \pm 3.3$ | $63.3 \pm 1.7$ | $49.4 \pm 0.9$ | $21.7 \pm 0.4$ |

In Table 2, we provide the maximum time consumption for a single round of FedTrans and other baselines across 300 communication rounds of Federated Learning (FL) communication. Additionally, it includes cumulative time consumption when reaching the target accuracy of 63% (i.e., the maximum accuracy achievable by all methods). The results reveal that, although FedTrans requires more time for each round compared to the baselines, the overall time consumption to reach the target accuracy with our proposed FedTrans is the shortest, showing an impressive speedup of more than 56%. This is primarily because FedTrans requires fewer rounds to attain the targeted accuracy.

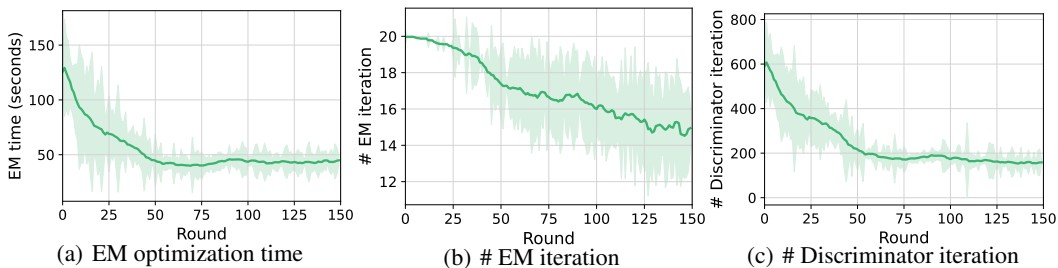

(a) EM optimization time     (b) # EM iteration     (c) # Discriminator iteration

Figure 12: The varying optimization overhead in FedTrans across communication round. From left to right: total optimization time, total EM iterations, and the Discriminator iteration required in variational inference.

Furthermore, we also explore the optimization overhead of the variational EM algorithm that changes as FL proceeds. Figure 12 shows the EM optimization time, EM iterations, and Discriminator (i.e., M-step) iterations varying with communication rounds. The overall optimization time significantly decreases as FL progresses, attributed to the diminishing trend observed in both EM iterations and discriminator iterations. Moreover, the time cost of training the discriminator takes a larger proportion of the total optimization time. As a result, the overall optimization time across communication rounds exhibits a similar trend as the discriminator iterations.

# F  EVALUATION ON PRACTICAL RESOURCE-CONSTRAINED TESTBED

We also build a testbed consisting of 21 Raspberry Pi 4B embedded devices, as shown in Figure 13(a), to evaluate the performance of FedTrans in practical resource-constrained scenarios.

**HAR Dataset.** We conduct the experiments on the Human Activity Recognition (HAR) dataset collected from onboard inertial sensors (accelerometer and gyroscope) for distinguishing six activities of daily living: *walking, walking upstairs/downstairs, sitting, standing, and laying.* HAR dataset shows inherent suitability for the FL scenarios since the data is naturally split by different participants. To be more specific, HAR contains data collected from 30 different users, and we take 21 users as the training data and augment the number of clients from 21 to 42 by assigning the data of each user to two clients. We run the FedTrans on a testbed and each Raspberry Pi simulates

Figure 13: Evaluation of FedTrans on the HAR dataset implemented on resource-constrained devices, across three settings of client flip rate ($\epsilon_c = 0.2, 0.3, 0.5$). **Our built testbed: 21 Raspberry Pi 4B embedded devices connect to an FL server wirelessly via a TP-Link WiFi Router.**

two clients and connects to the laptop wirelessly through a TP-Link WiFi Router. Here, we employ a shallow CNN with two convolutional layers for the HAR task and only consider the *Random Flipping* noise with three different proportions of corrupted clients ranging from 30% to 50%.

Results in Figure 13 show FedTrans can already outperform FedAvg by a margin from the beginning. We notice that the gap becomes larger when the number of corrupted clients increases. While FedAvg performance increases when the training proceeds to later rounds, it remains lower than FedTrans. This shows the advantage of selecting clean clients even though the number of much fewer than the clients selected by FedAvg.

**Energy-to-accuracy.** We report the energy consumption (measured by the power meter) when the global model achieves the same performance (i.e., the best accuracy of FedAvg) utilizing the CIFAR10 dataset on our testbed shown in Figure 9(a). FedTrans is more energy-efficient even with the extra overhead at the server: it reduces the energy consumption by up to 66.1%. This is mainly due to the fewer training rounds required when FL is training with FedTrans: it requires only 52.2%, 79.2%, and 74.1% rounds to reach FedAvg's best performance for three client noise ratios, respectively.

# G    PERFORMANCE ON OTHER DATA DISTRIBUTIONS

**IID setting.** We conduct experiments under the IID setting using the CIFAR10 and FMNIST dataset, evaluating the performance of baselines in mixed noise scenarios, The results are presented in Table 3 and Table 4. The data reported in the tables is based on five trials.

Table 3: Global model accuracy under six types of noise configurations using CIFAR10 dataset. Distribution of the local data is followed by IID setting, and $\epsilon = 30\%$ of participating 100 clients are corrupted.

| | CIFAR-10, MobileNetV2, *IID* | | | | | |
|---|---|---|---|---|---|---|
| | **Hybrid** (*across-*) | **Hybrid** (*intra-*) | **Label** (*across-*) | **Label** (*intra-*) | **Image** (*across-*) | **Image** (*intra-*) |
| FedAvg (McMahan et al., 2017) | 82.2% ± 0.1% | 80.2% ± 0.2% | 81.6% ± 0.1% | 81.3% ± 0.0% | 82.5% ± 0.0% | 81.6% ± 0.1% |
| FLDebugger (Li et al., 2021) | 82.1% ± 0.0% | 81.5% ± 0.3% | 82.0% ± 0.2% | 81.4% ± 0.4% | 81.9% ± 0.1% | 82.2% ± 0.2% |
| Oort (Lai et al., 2021) | 66.7% ± 0.4% | 63.4% ± 0.5% | 64.7% ± 0.7% | 64.8% ± 0.0% | 70.2% ± 0.3% | 69.2% ± 0.5% |
| Robust-FL (Yang et al., 2022b) | 78.4% ± 0.0% | 77.5% ± 0.6% | 77.9% ± 0.1% | 78.2% ± 0.4% | 78.0% ± 0.0% | 76.9% ± 0.2% |
| RHFL (Fang & Ye, 2022) | 83.1% ± 0.2% | 82.3% ± 0.1% | 83.4% ± 0.1% | 83.4% ± 0.2% | 82.9% ± 0.5% | 82.1% ± 0.1% |
| DivFL (Balakrishnan et al., 2022) | 79.4% ± 0.1% | 78.4% ± 0.2% | 77.5% ± 0.2% | 77.6% ± 0.0% | 80.1% ± 0.0% | 78.7% ± 0.2% |
| FedCorr (Xu et al., 2022) | 79.1% ± 0.0% | 79.6% ± 0.1% | 78.7% ± 0.1% | 78.3% ± 0.1% | 79.8% ± 0.3% | 78.9% ± 0.2% |
| **FedTrans** | 83.9% ± 0.3% | 83.8% ± 0.1% | 84.2% ± 0.2% | 84.0% ± 0.0% | 84.1% ± 0.1% | 83.8% ± 0.2% |

Table 4: Global model accuracy under IID setting with varying ratios of the corrupted clients using FMNIST dataset.

| | The Ratio of Corrupted Clients | | | | |
|---|---|---|---|---|---|
| | **30%** | **40%** | **50%** | **60%** | **70%** |
| FedAvg w/o noise (McMahan et al., 2017) | 91.2% ± 0.0% | 91.2% ± 0.0% | 91.2% ± 0.0% | 91.2% ± 0.0% | 91.2% ± 0.0% |
| FedAvg w/ noise (McMahan et al., 2017) | 90.4% ± 0.2% | 90.3% ± 0.1% | 90.0% ± 0.1% | 89.7% ± 0.1% | 87.6% ± 0.0% |
| **FedTrans** | 90.5% ± 0.0% | 90.9% ± 0.1% | 90.3% ± 0.1% | 90.4% ± 0.7% | 90.2% ± 0.0% |

Results in these two tables demonstrate that the negative of unreliable clients could be exacerbated by the complexity of the task. Even with mixed noise, a significant accuracy drop only becomes discernible when the ratio of corrupted clients surpasses 60% in the relatively straightforward task (i.e., FMNIST dataset). FedAvg suffers from significant performance degradation under mixed noise on the complex task (i.e., CIFAR10 dataset), while FedTrans outperforms other baselines in effectively mitigating the effects of such complex noise.

**H2C setting.** We also explored a more severe non-IID experimental setting where each client holds only two classes of data, denoted as H2C. Table 5 reports the results of the compared methods in different noise configurations under 30% corrupted clients.

Table 5: Global model accuracy under six types of noise configurations. Distribution of the local data is followed by H2C setting, and $\epsilon = 30\%$ of participating 100 clients are corrupted.

| | **Hybrid** (*across-*) | **Hybrid** (*intra-*) | **Label** (*across-*) | **Label** (*intra-*) | **Image** (*across-*) | **Image** (*intra-*) |
|---|---|---|---|---|---|---|
| CIFAR-10, MobileNetV2, *H2C* | | | | | | |
| FedAvg (McMahan et al., 2017) | 43.6% ± 0.1% | 37.9% ± 0.3% | 43.1% ± 0.3% | 40.3% ± 0.5% | 43.5% ± 0.2% | 44.9% ± 0.5% |
| FLDebugger (Li et al., 2021) | 37.7% ± 0.1% | 40.7% ± 0.7% | 37.1% ± 0.7% | 37.7% ± 0.2% | 38.2% ± 0.9% | 40.2% ± 1.0% |
| Oort (Lai et al., 2021) | 33.2% ± 0.2% | 18.7% ± 0.7% | 34.9% ± 0.4% | 28.3% ± 0.7% | 44.6% ± 0.8% | 41.1% ± 0.1% |
| Robust-FL (Yang et al., 2022b) | 44.4% ± 0.4% | 36.1% ± 0.3% | 44.3% ± 0.3% | 43.1% ± 0.4% | 46.3% ± 0.5% | 43.9% ± 1.0% |
| RHFL (Fang & Ye, 2022) | 43.8% ± 0.5% | 36.0% ± 0.8% | 46.6% ± 1.1% | 43.7% ± 0.8% | 44.9% ± 0.8% | 43.2% ± 0.3% |
| DivFL (Balakrishnan et al., 2022) | 45.1% ± 0.9% | 40.4% ± 0.4% | 42.64% ± 0.2% | 41.3% ± 0.0% | 44.1% ± 0.1% | 45.4% ± 0.2% |
| FedCorr (Xu et al., 2022) | 48.0% ± 0.5% | 32.1% ± 0.1% | 47.7% ± 0.3% | 45.0% ± 0.3% | 40.1% ± 1.1% | 40.4% ± 0.3% |
| **FedTrans** | 50.0% ± 0.7% | 48.4% ± 0.6% | 45.3% ± 0.9% | 48.8% ± 0.7% | 46.6% ± 0.6% | 46.4% ± 0.8% |
| FMNIST, LeNet-5, *H2C* | | | | | | |
| FedAvg (McMahan et al., 2017) | 76.1% ± 0.1% | 75.7% ± 0.1% | 76.0% ± 0.5% | 75.7% ± 0.6% | 76.7% ± 0.2% | 76.9% ± 0.2% |
| FLDebugger (Li et al., 2021) | 68.2% ± 0.2% | 69.3% ± 0.2% | 72.1% ± 0.1% | 69.0% ± 0.3% | 57.9% ± 0.2% | 56.8% ± 0.2% |
| Oort (Lai et al., 2021) | 72.3% ± 0.3% | 48.0% ± 0.7% | 66.2% ± 0.9% | 45.7% ± 0.8% | 78.9% ± 0.1% | 78.1% ± 0.8% |
| Robust-FL (Yang et al., 2022b) | 79.8% ± 0.2% | 79.9% ± 0.0% | 79.8% ± 0.8% | 78.3% ± 0.2% | 79.3% ± 0.3% | 80.3% ± 0.4% |
| RHFL (Fang & Ye, 2022) | 80.7% ± 0.1% | 81.2% ± 0.2% | 80.6% ± 0.2% | 80.0% ± 0.3% | 80.8% ± 0.2% | 80.9% ± 0.2% |
| DivFL (Balakrishnan et al., 2022) | 76.8% ± 0.7% | 76.4% ± 0.9% | 75.1% ± 0.4% | 75.2% ± 0.2% | 76.0% ± 0.3% | 77.3% ± 0.5% |
| FedCorr (Xu et al., 2022) | 82.1% ± 0.1% | 81.7% ± 0.1% | 82.2% ± 0.4% | 81.3% ± 0.1% | 79.3% ± 0.2% | 79.3% ± 0.2% |
| **FedTrans** | 82.4% ± 0.2% | 84.2% ± 0.6% | 84.3% ± 0.1% | 84.7% ± 0.3% | 83.5% ± 0.2% | 83.2% ± 0.2% |

