# OpenReview forum: "FedTrans: Client-Transparent Utility Estimation for Robust Federated Learning"
_ICLR.cc/2024/Conference — ICLR 2024 poster_

### Official Review · Reviewer_PLVU · 2023-10-26

**Soundness:** 3 good
**Presentation:** 3 good
**Contribution:** 3 good
**Rating:** 6
**Confidence:** 3

**Summary:**

* This paper proposes a Bayesian framework to estimate the client utility on different levels of noisy data.

* Two metrics, i.e., weight-based utility estimation, and performance-based utilization estimation are applied into the Bayesian framework.
For the first metric, last-layer weights are trained on a server-owned clean auxiliary dataset/and the noisy dataset is provided as a label. For the second metric, the inference results over the selected model in the auxilliary dataset are used for labeling.

**Strengths:**

* The paper is very well-written and the motivation for proposing a utility estimation method is clear.

* The addressed problem is important and addresses the pain in the deployment of federated learning.

* The method looks technically solid, and the formulation/description is concise but rigorous, but I couldn't check their correctness because of a lack of background in the Bayesian framework.

* The experiment is also comprehensive, which demonstrates the effectiveness of the utility estimation. Ablation study and analysis of different simulation settings are also provided.

**Weaknesses:**

* The Bayesian inference part is difficult to read through for readers without related knowledge. The authors may want to  introduce the framework of EM updates in the Appendix for readers without such background,

* The authors may discuss and contrast their Bayesian-based solution and multi-arm bandit-based client selection framework,  (Lai et al,2021), (Huang et al,2020), (Xia et al,2020)， as both frameworks aim to balance exploration and exploitation by providing some label signal for the client selection process.


* Minor: the margin of the headers of Section 4 and Section 5 are modified. It is suggested the authors obey the author's guidelines and keep the original format.


Lai F, Zhu X, Madhyastha H V, et al. Oort: Efficient federated learning via guided participant selection[C]//15th {USENIX} Symposium on Operating Systems Design and Implementation ({OSDI} 21). 2021: 19-35.

Huang T, Lin W, Wu W, et al. An efficiency-boosting client selection scheme for federated learning with fairness guarantee[J]. IEEE Transactions on Parallel and Distributed Systems, 2020, 32(7): 1552-1564.

Xia W, Quek T Q S, Guo K, et al. Multi-armed bandit-based client scheduling for federated learning[J]. IEEE Transactions on Wireless Communications, 2020, 19(11): 7108-7123.

**Questions:**

As I am not so familiar with the EM framework, I am wondering why the weight-based utility estimation is only involved in the M step, while the performance utilization estimation is involved in the E step. Is that because the performance-based utility label is binary?

It is shown on page 6 that Algorithm 1 is run in each round of the federated learning process. Will the discriminator weight be inherited from that obtained in the previous round? Will utility inference become more accurate when rounds in the federated learning process increase? If so, can the authors demonstrate how the utility curve evolves with the rounds go?

---

> ### Author Response · Authors · 2023-11-20
> **Response to reviewer 6FPe (Part 1)**
>
> We thank the reviewer for the insightful comments. Below are our responses.
>
> `[1/5] The Bayesian inference part is difficult to read through for readers without related knowledge. The authors may want to introduce the framework of EM updates in the Appendix for readers without such background.`
>
> We thank the reviewer for pointing this out. We have updated Appendix A.1 by briefly introducing the basic idea for the EM algorithm. For a more extensive introduction to variational EM, a good reference could be Tzikas et al. 2008; Blei et al. 2017 provide a comprehensive review of the topic.
>
> [1] Tzikas, Dimitris G., Aristidis C. Likas, and Nikolaos P. Galatsanos. "The variational approximation for Bayesian inference." IEEE Signal Processing Magazine 25.6 (2008): 131-146.
>
> [2] Blei, David M., Alp Kucukelbir, and Jon D. McAuliffe. "Variational inference: A review for statisticians." Journal of the American statistical Association 112.518 (2017): 859-877.
>
> `[2/5] The authors may discuss and contrast their Bayesian-based solution and multi-arm bandit-based client selection framework, (Lai et al,2021), (Huang et al,2020), (Xia et al,2020)， as both frameworks aim to balance exploration and exploitation by providing some label signal for the client selection process.`
>
> We thank the reviewer for mentioning these papers. FedTrans primarily focuses on Bayesian-based parameter estimation, distinguishing itself from the exploration-exploitation client searching emphasized in the mentioned works. The distinction in approaches between FedTrans and these papers is rooted in the diverse interpretations of client selection. Specifically, the three papers address the optimal selection of participating clients in each communication round, while FedTrans addresses the problem of selective aggregation of local updates from random clients participating in each communication round.
>
> The different goals have direct implications for the methods in the existing literature and in our work. In selecting participating clients, existing methods calculate relevant measures of clients (e.g., the amount of time taken in each round, or training loss) depending on the optimization objectives (e.g., low training latency, high global model accuracy, etc.) they prioritize. With those measures, it is a natural choice to frame the problem as a multi-arm bandit problem to search for the participating clients. In the selective aggregation of clients' updates that we address in our work, client utility cannot be calculated directly but needs to be modeled and inferred since it depends on local data quality. That is the very reason why we adopted a Bayesian framework. Our work, therefore, frames the problem as utility inference based on observations, such as local updates and local model performance recorded in the round reputation matrix $\textbf{R}$.
>
> `[3/5] Minor: the margin of the headers of Section 4 and Section 5 are modified. It is suggested the authors obey the author's guidelines and keep the original format.`
>
> Thank you for pointing this out. We have revised the paper by sticking to the original format.
>
> `[4/5] As I am not so familiar with the EM framework, I am wondering why the weight-based utility estimation is only involved in the M step, while the performance utilization estimation is involved in the E step. Is that because the performance-based utility label is binary?`
>
> We clarify that the optimization of the two modules, weight-based estimation and performance-based estimation, does not directly align with the M-step and E-step, respectively. The E-step updates the two latent variables in both two modules, and the M-step updates the discriminator in the weight-based estimation. Specifically, the E-step iteratively updates the (distribution parameters of) latent variables (single circled nodes in Figure 3), i.e. the selection decision $s$ in the weigh-based module and informativeness $r$ of round-reputation matrix $\mathbf{R}$ in the performance-based module. In the subsequent M-step, we use the updated $s$ as the label to iteratively update the learning-based discriminator (i.e., training a neural network via back-propagation). Therefore, the logic behind the system design is that we first propose two estimation strategies and formulate a Bayesian framework to integrate them. We then employ variational EM to optimize variables in the Bayesian framework, wherein the E-step and M-step are designed to iteratively update the variables related to both the two modules.

---

> ### Author Response · Authors · 2023-11-20
> **Response to reviewer 6FPe (Part 2)**
>
> `[5/5] It is shown on page 6 that Algorithm 1 is run in each round of the federated learning process. Will the discriminator weight be inherited from that obtained in the previous round? Will utility inference become more accurate when rounds in the federated learning process increase? If so, can the authors demonstrate how the utility curve evolves with the rounds go?`
>
> The discriminator indeed inherits weight parameters from the previous round. In each communication round, the learning-based discriminator is initialized with the updated weight from the EM step in the last communication round. The underlying rationale for inheriting weights lies in the incremental samples for updating the learning-based discriminator: more local updates become available over the communication rounds, thus providing more training instances to the learning-based discriminator, allowing it to improve over time; it is therefore, beneficial to inherit weights from the previous communication rounds.
>
> Such an inheriting strategy offers benefits to both inference performance and algorithm efficiency. On the one hand, with the FL proceeding, the inference of client utility becomes more accurate, as shown in the updated Figure 11 in Appendix C. Here, we assess the utility estimation by measuring the strength of the correlation between the inferred client utility and its actual local noise rate. On the other hand, the convergence of discriminator training (i.e., M-step) speeds up with the communication rounds, as shown in the updated Figure 12 (b), resulting in the reduction in the computational overhead of FedTrans in the long run.

---

> > ### Comment · Reviewer_PLVU · 2023-11-22
> > **Reply to authors**
> >
> > Thanks for the rebuttal.  I would keep my score, which I believe is fair.

---

> > > ### Author Response · Authors · 2023-11-22
> > > **Thank you for your reply**
> > >
> > > Dear Reviewer PLVU,
> > >
> > > Thank you for taking the time to review our rebuttal and for acknowledging the supplementary experimental results we provided. We appreciate your insightful review to enhance the quality of our work.
> > >
> > > Best,
> > > Authors

---

### Official Review · Reviewer_kySo · 2023-10-29

**Soundness:** 2 fair
**Presentation:** 2 fair
**Contribution:** 2 fair
**Rating:** 6
**Confidence:** 3

**Summary:**

In this paper, the authors propose a novel Bayesian method designed to achieve robust aggregation on the server side within the framework of federated learning, effectively addressing the challenges posed by heterogeneous and noisy data. Central to this methodology is the use of a small yet pristine and balanced dataset, which resides on the server side and plays a crucial role in approximating the utility of each client. The experiment results validate the efficacy of the proposed approach, showcasing its potential practical benefits.

**Strengths:**

1. The paper addresses a significant issue in the federated learning (FL) environment, presenting a clear and well-founded motivation.

2. The clarity and simplicity of the writing style make the content accessible and easy to understand.

**Weaknesses:**

1. The novelty of this paper appears somewhat limited. There have been extensive prior studies on federated learning focused on client utility. It would be beneficial if the author could provide further clarification regarding the unique contributions of this work.

2. The proposed method appears to have a high dependence on the server dataset, which significantly limits its potential use cases. This limitation substantially reduces the generality of the proposed method.

3. While the authors do provide a convergence analysis in the appendix, it lacks a proper derivation of the convergence rate. In comparison to random sampling, the theoretical advantages of the proposed method remain unknown.

4. The proposed method introduces additional computational overhead per round, which could potentially increase the time required for each round when compared to the baseline methods.

**Questions:**

See the weakness above.

---

> ### Author Response · Authors · 2023-11-20
> **Response to reviewer kySo (Part 1)**
>
> We thank the reviewer for the insightful comments. Below are our responses.
>
> `[1/4] The novelty of this paper appears somewhat limited. There have been extensive prior studies on federated learning focused on client utility. It would be beneficial if the author could provide further clarification regarding the unique contributions of this work.`
>
> The purpose of client utility estimation varies in existing works. Some works [1, 2] infer the client utility in response to challenges arising from heterogeneous data distribution and systems. It's crucial to note that the term *statistical utility* in these papers typically pertains to the local data distribution rather than the local data quality that our paper focuses on. In the context of noisy clients, some approaches rely on comparing local weight parameters across clients, which may have poor performance when detecting the noisy client, while others lack transparency, requiring clients to report specific information to the central server.
>
> Our novel contribution lies in the introduction of a transparent client utility estimation method, i.e., the Bayesian framework and the corresponding variational inference algorithm. Our method maintains the same level of privacy constraints as FedAvg since the local data and the training process are unchanged from the client's perspective. Furthermore, the method is compatible with other SOTA baselines since it works as an independent module on the server side.
>
> [1] "Oort: Efficient federated learning via guided participant selection." 15th USENIX Symposium on Operating Systems Design and Implementation (OSDI 21). 2021.
>
> [2] "PyramidFL: A fine-grained client selection framework for efficient federated learning." Proceedings of the 28th Annual International Conference on Mobile Computing And Networking (MobiCom 22). 2022.
>
> `[2/4] The proposed method appears to have a high dependence on the server dataset, which significantly limits its potential use cases. This limitation substantially reduces the generality of the proposed method.`
>
> We want to highlight that FedTrans is not the first paper to use an auxiliary dataset. Some previous works [1, 2, 3] also introduce additional data to improve the performance of Federated Learning. Although the auxiliary dataset comes at a cost, it is affordable for the server because the size of the auxiliary dataset is very small. For instance, in our experiments, we construct the auxiliary dataset by randomly selecting only 200 samples from the test set of CIFAR10 and FMNIST, respectively (note that the accuracy we report in the paper is evaluated on the rest of samples in the test set). Thus the size of the auxiliary dataset is very small compared to the training dataset used for clients: in a proportion of only 0.4\% and 0.3\%, respectively. We note that in the practical deployment, it is feasible for the service provider to obtain a small amount of auxiliary dataset (e.g., by soliciting data from paid, anonymous workers).
>
> Furthermore, our empirical evaluation underscores the cost-effectiveness of our approach. Figure 7 shows that FedTrans effectively estimates client utility with only a small amount of auxiliary data. Notably, there is still a significant effect on the global model when the auxiliary dataset contains a mere 50 data instances (0.1\% of the training dataset for CIFAR10), which signifies the cost-efficiency of FedTrans in making use of auxiliary data.
>
> [1] "Communication-efficient on-device machine learning: Federated distillation and augmentation under non-iid private data." Workshop on Machine Learning on the Phone and Other Consumer Devices (in Conjunction with NeurIPS 2018).
>
> [2] "Overcoming noisy and irrelevant data in federated learning." 2020 25th International Conference on Pattern Recognition (ICPR). IEEE, 2021.
>
> [3] "With a Little Help from My Friend: Server-Aided Federated Learning with Partial Client Participation." Workshop on Federated Learning: Recent Advances and New Challenges (in Conjunction with NeurIPS 2022).
>
> `[3/4] While the authors do provide a convergence analysis in the appendix, it lacks a proper derivation of the convergence rate. In comparison to random sampling, the theoretical advantages of the proposed method remain unknown.`
>
> We appreciate the reviewer for pointing it out. We have updated the discussion on the convergence rate in Appendix B.

---

> ### Author Response · Authors · 2023-11-20
> **Response to reviewer kySo (Part 2)**
>
> `[4/4] The proposed method introduces additional computational overhead per round, which could potentially increase the time required for each round when compared to the baseline methods.`
>
> Indeed, the utility inference in FedTrans introduces additional computational overhead. While FedTrans takes a longer time in each communication round, the removal of corrupted updates for aggregation benefits global model convergence in the long run, thereby FedTrans exhibits superior time-to-accuracy in the presence of noisy updates.
>
> As shown in Table 2 in Appendix E, the overall time consumption for reaching the target accuracy showcases an impressive speedup with FedTrans compared to other baseline methods. We also measure the energy consumption of FedTrans and the baseline method on a testbed consisting of 21 Raspberry Pi 4B. As shown in Appendix F, FedTrans is more energy-efficient even with the extra overhead at the server. The results highlight the time-to-accuracy performance of FedTrans, which is primarily attributed to requiring fewer rounds to attain the targeted accuracy.
>
> Moreover, we present new experimental results, as shown in Figure 12 in Appendix E, on the overall optimization overhead in FedTrans varying across communication rounds. The results reveal the diminishing overhead imposed by FedTrans on the server as FL proceeds, facilitated by the EM algorithm becoming easier to converge.

---

> ### Author Response · Authors · 2023-11-22
> **Look Forward to Your Response**
>
> Dear Reviewer kySo,
>
> As the discussion period nears its conclusion, we wish to ensure that all your concerns have been adequately addressed. We would greatly appreciate any additional follow-up clarifications or questions you may have.
>
> Thank you,
> Authors

---

> ### Author Response · Authors · 2023-11-23
> **Gentle reminder for response**
>
> Dear Reviewer kySo,
>
> Thank you very much for your time and effort in coordinating the review of our paper. As the discussion deadline approaches, we hope our responses address all your concerns. We eagerly anticipate feedback from you.
>
> Thank you, Authors

---

### Official Review · Reviewer_DCr1 · 2023-10-31

**Soundness:** 3 good
**Presentation:** 2 fair
**Contribution:** 2 fair
**Rating:** 6
**Confidence:** 3

**Summary:**

The paper considers the problem of performance degradation caused by the presence of noisy clients in federated learning. In order to mitigate this degradation, the paper proposes a client selection policy based on  FedTrans, a Bayesian framework for client utility estimation. The paper constructs a probabilistic graphic model determining the relationship between the client utility, the round reputation matrix, the round informativeness, and some other parameters of the problem (e.g., clients models weights, discriminator model weights, and prior distributions parameters). The paper uses Variational Expectation Maximization to infer the parameters of this probabilistic graphical model. Finally, the paper conducts numerical experiments on FEMNIST and CIFAR-10 datasets with different types and levels of noise. The numerical experiments lead to the following conclusions: 1) FedTrans outperforms other methods. 2) Other methods cannot take advantage for small auxiliary dataset, as opposed to FedTrans. 3) Combining both the round-reputation matrix and the discriminator model is crucial for FedTrans to achieve good performance. 4) The performance of FedTrans is robust to variation in the size of the auxiliary dataset, however, the performance drops if the auxiliary dataset is scarce.

**Strengths:**

- The paper effectively motivates the problem of performance degradation under the presence of noisy clients.
- The paper clearly justifies the feasibility of accessing an auxiliary dataset.
- The definition of the probabilistic graphical model, and the execution of Variational EM are overall correct.
- The proposed method does not require the clients to perform any additional computation.
- The numerical experiments, although being restricted to only two datasets, are fair, and evaluate all the important aspects of the proposed approach.

**Weaknesses:**

- The paper relies on the availability of an auxiliary public dataset at the server.
- The clarity of the probabilistic graphical model explanation is lacking, and the rationale behind the modeling choices is not consistently elaborated upon.
    -  It is unclear what "the top-layer" means.
    - $x_j$ is obtained using $W_{i, j}$.  It raises the question of why the paper opts not to employ $x_{i, j}$ instead.
    - Further clarification is needed to justify (5).
- I am surprised by the drop of DivFL and FedCorr after 2the fine-tuning. My guess is that the fine-tuning employed a large learning rate, or a large number of epochs.
- Figure 6 highlights a limitation of the proposed method: relying solely on the round-reputation or the discriminator model results in inferior performance compared to alternative methods. It is plausible that the superior performance of FedTrans stems from its stacking of two learning methods.
- The evaluation criteria is unclear. Are the reported results evaluated only on "good" (i.e., mot noisy) clients, or on all the clients.
- Similar to most robust federated learning approaches, FedTrans might raise fairness concerns, as it may eliminate benign clients from the training if their local data distribution is very different from the majority of clients.
- Other minor issues:
    - The conclusions of Figure 1 might be due to a poor choice of the learning rate.
    - I am not sure that Figure 1 brings any significant value to the paper. My understanding is that the main conclusion obtained from Figure is that it is important to select reliable clients for training the global model. This conclusion is obvious.
    - In the first sentence in Section 1; lives -> live.
    - what is the meaning of "without loss of generality" in Section 1. Does it mean that the same results hold if we vary the flip rate?
    - As opposed to the paper claim, I think that the calibration/computation of the weights $p_j$ in (1) is crucial. Please, note that $p_j$ in (1) is different from $p_j$ used in Appendix B.
    - I think that $\Theta$ is missing in the probabilistic graphical model depicted in Figure 3.

**Questions:**

- Could you, please, discuss the fairness implication of FedTrans?
- Could you, please, clarify if the evaluation r2eports the performance of benign clients only, or includes the performance of noisy clients?
- Could you, please, report the success rate of FedTrans in detecting the noisy clients?

---

> ### Author Response · Authors · 2023-11-20
> **Response to reviewer DCr1 (Part 1)**
>
> We thank the reviewer for the insightful comments. Below are our responses.
>
> `[1/10] The paper relies on the availability of an auxiliary public dataset at the server.`
>
> We want to highlight that FedTrans is not the first paper to use an auxiliary dataset. Some previous works [1, 2, 3] also introduce additional data to improve the performance of Federated Learning. Although the auxiliary dataset comes at a cost, it is affordable for the server because the size of the auxiliary dataset is very small. For instance, for our experiments, we construct the auxiliary dataset by randomly selecting only 200 samples from the test set of CIFAR10 and FMNIST, respectively (note that the accuracy we report in the paper is evaluated on the rest of the samples in the test set). Thus the size of the auxiliary dataset is very small compared to the training dataset used for clients: in a proportion of only 0.4\% and 0.3\%, respectively. We note that in the practical deployment, it is feasible for the service provider to obtain a small amount of auxiliary dataset (e.g., by soliciting data from paid, anonymous workers).
>
> [1] "Communication-efficient on-device machine learning: Federated distillation and augmentation under non-iid private data." Workshop on Machine Learning on the Phone and Other Consumer Devices (in Conjunction with NeurIPS 2018).
>
> [2] "Overcoming noisy and irrelevant data in federated learning."  25th IEEE International Conference on Pattern Recognition (ICPR), 2021.
>
> [3] "With a Little Help from My Friend: Server-Aided Federated Learning with Partial Client Participation." Workshop on Federated Learning: Recent Advances and New Challenges (in Conjunction with NeurIPS 2022).
>
> `[2/10] The clarity of the probabilistic graphical model explanation is lacking, and the rationale behind the modeling choices is not consistently elaborated upon.`
>
> **Q: It is unclear what "the top-layer" means.**
>
> The 'top-layer', wherein the weights are the input of the discriminator, is the last classification layer before the softmax of the local model. Weights in the topmost layer are the most discriminative for our utility inference since they are most relevant for the given task [1].
>
> [1] "Fedrs: Federated learning with restricted softmax for label distribution non-iid data." Proceedings of the 27th ACM SIGKDD Conference on Knowledge Discovery & Data Mining. 2021.
>
> **Q: $x_ {j}$ is obtained using $W_ {i,j}$. It raises the question of why the paper opts not to employ $x_{i,j}$ instead.**
>
> We thank the reviewer for pointing this out. $x_ {j}$ is extracted from the weight parameters of the $j$-th client's local model, varying with communication rounds. It is indeed a more explicit notation $x_ {i,j}$. However, client $j$ may not participate in every communication round. We intentionally opted for the notation $x_ {j}$ while omitting the round index $i$ to emphasize the general applicability of our approach across communication rounds.
>
> **Q: Further clarification is needed to justify (5).**
>
> Equation (5) describes the relationship between two latent variables (i.e., selection decision $s_ i$ and round informativeness $r_ i$) and observed variables (i.e., entries in the round reputation matrix $\mathbf{R}$). The reason behind Equation (5) is that the informativeness of $i$-th round indicates the probability if estimated client performance $s_ j$ matches its actual performance $\mathbf{R} _{i, j}$, for all the clients in the $i$-th round (i.e., $\forall j \in \mathcal{J} ^{i}$).

---

> ### Author Response · Authors · 2023-11-20
> **Response to reviewer DCr1 (Part 2)**
>
> `[3/10] I am surprised by the drop of DivFL and FedCorr after the fine-tuning. My guess is that the fine-tuning employed a large learning rate, or a large number of epochs.`
>
> We understand the reviewer's concern. In response, we provide more detailed experimental results regarding two fine-tuning methods, as shown in the table below (hybrid (across) noise type). These results consider five distinct fine-tuning epochs across two different learning rates under two datasets.
>
> |||||hybrid (across-)|||
> |-|-|-|-|-|-|-|
> |Dataset|Epoch||Learning rate=0.001||Learning rate=0.0001||
> ||||Fine-tuned FedCorr|Fine-tuned DivFL|Fine-tuned FedCorr|Fine-tuned DivFL|
> |Cifar10|1||69.6%±0.3%|69.8%±0.2%|69.3%±0.2%|67.7%±0.4%|
> ||2||70.0%±0.2%|69.8%±0.3%|68.7%±0.2%|68.0%±0.3%|
> ||3||70.6%±0.2%|70.3%±0.3%|69.3%±0.3%|68.8%±0.4%|
> ||4||71.1%±0.3%|70.6%±0.3%|69.4%±0.3%|68.0%±0.3%|
> ||5||70.7%±0.2%|68.7%±0.1%|70.1%±0.2%|69.0%±0.7%|
> |FMnist|1||84.4%±0.4%|84.3%±0.1%|83.8%±0.1%|83.6%±0.3%|
> ||2||84.9%±0.1%|84.3%±0.3%|83.6%±0.2%|83.9%±0.4%|
> ||3||84.4%±0.2%| 84.1%±0.2%|84.4%±0.2%|83.5%±0.2%|
> ||4||84.7%±0.1%|84.3%±0.2%|84.6%±0.3%|83.4%±0.1%|
> ||5||84.6%±0.1%| 84.2%±0.1%|84.3%±0.1%|83.8%±0.2%|
>
> We observe a consistent accuracy drop in FedCorr and DivFL, which is likely due to the limited size of the auxiliary dataset used for fine-tuning that causes overfitting of the global model. **This signifies the non-trivial challenge of effectively utilizing auxiliary datasets**, which is the very problem our proposed FedTrans addresses.
>
> The original version of the paper reported results considering only two different fine-tuning epochs (i.e., 1 and 2 epochs) with a 0.001 learning rate. In Table 1 of the revised paper, we have updated the latest results by selecting the best accuracy of these two comparison methods from all configurations.
>
> `[4/10] Figure 6 highlights a limitation of the proposed method: relying solely on the round reputation or the discriminator model results in inferior performance compared to alternative methods. It is plausible that the superior performance of FedTrans stems from its stacking of two learning methods.`
>
> The performance-based and weight-based utility inference methods, corresponding to the round reputation matrix and discriminator, respectively, are inherently complementary. This synergy forms the basis of the Bayesian framework in FedTrans, where these two methods are integrated to achieve high efficacy of client utility estimation. On the one hand, the weight-based discriminator is trained to discriminate clients, leveraging synthetic clients constructed using the auxiliary data; it is thus limited to the number of synthetic clients and the way noise is introduced. On the other hand, the performance-based method uses individual data instances in the auxiliary data for client performance evaluation, which provides an alternative way of leveraging the auxiliary data. Our Bayesian framework considers both the clients' historical performance and predictions from the discriminator inherited across rounds, to update the client utility iteratively and incrementally.
>
> `[5/10] The evaluation criteria are unclear. Are the reported results evaluated only on "good" (i.e., not noisy) clients, or on all the clients?`
>
> In this paper, we propose FedTrans to infer the client utility in the context of existing corrupted clients with local noise. Therefore, we report the experimental results of the global model performance when a proportion of $\epsilon$ corrupted clients involved in the FL training process. Guided by the estimated utility from FedTrans, the global model selectively aggregates the updates from clients with high utilities.
>
> Figure 8 suggests that involving less noisy updates in server aggregation has the potential to enhance global model accuracy while maintaining the convergence speed. Our experiments employ a threshold-based selection strategy, excluding the updates from the clients with the utility $\theta$ below 0.5, for the global model aggregation. While we focus on utility estimation, we clarify that client selection is not the primary focus of this paper. Given the inversely proportional relation between utilities and local noise rates (Figure 4), the estimated utility can be also used for selecting a certain number of clients with the best utilities.

---

> ### Author Response · Authors · 2023-11-20
> **Response to reviewer DCr1 (Part 3)**
>
> `[6/10] Similar to most robust federated learning approaches, FedTrans might raise fairness concerns, as it may eliminate benign clients from the training if their local data distribution is very different from the majority of clients.`
>
> We appreciate the reviewer’s feedback on pointing out the fairness issue. We clarify that the focus of our work is estimating the utility of the clients affected by data noise. Our experiments do not show evidence of the data distribution's effect on the estimated client utility. We acknowledge the importance of the fairness issue and recognize its complexity: the potential dependency on data distributions and other criteria related to the definition of 'fairness', which is inherently context-dependent. We believe that addressing the fairness issue requires very different approaches and therefore believe that it would be more suitable to leave the problem for future work.
>
> `[7/10] Other minor issues`
>
> **Q: The conclusions of Figure 1 might be due to a poor choice of the learning rate.**
>
> The learning rate used for obtaining the results in Figure 1 is $10^{-2}$. We also did experiments with other learning rates. In the revised paper, we have added the results of using the learning rates with $10^{-1}$, $10^{-3}$, and $10^{-4}$ in Figure 9 (Appendix C). Figure 1 indeed reports the best performance for all data configurations (across all learning rates). We observe a larger degradation of the model performance introduced by data noise in non-IID data compared to IID data across all learning rates ranging from $10^{-1}$ to $10^{-4}$. The validity of this conclusion is further supported by the performance of FedTrans and baseline methods on IID settings in Appendix G. In this context, the improvement of global model accuracy is limited, primarily due to the marginal accuracy drop in vanilla FedAvg exposed to noisy updates with IID local data across clients.
>
> **Q: I am not sure that Figure 1 brings any significant value to the paper. My understanding is that the main conclusion obtained from Figure is that it is important to select reliable clients for training the global model. This conclusion is obvious.**
>
> We use Figure 1 to highlight the negative impact of noisy updates on the performance of the global model. We aim to emphasize that this negative impact is intricately associated with local data distribution. Noisy updates have a more pronounced effect when the local data is non-IID. Therefore, in our paper, we study the client utility estimation mainly under the assumption of a Dirichlet distribution of local data among the clients. This choice makes our experiments align well with the practical scenarios in reality.
>
> **Q: In the first sentence in Section 1; lives $\rightarrow$ live.**
>
> Thank you for pointing out the typo. We have fixed it in the updated paper.
>
> **Q: What is the meaning of "without loss of generality" in Section 1? Does it mean that the same results hold if we vary the flip rate?**
>
> Thank you for pointing this out. For the "without loss of generality", we mean that the results also hold under other types of noise, as shown later in Section 3. Here in Section 1, Figure 1 only shows the results under the setting of random label flipping, but note that the corrupted clients have different label flipping rates, with an average rate of 16.5\%. In the updated paper, we have rephrased the sentence to make it less confusing.
>
> **Q: As opposed to the paper claim, I think that the calibration/computation of the weights $p_ {j}$ in (1) is crucial. Please, note that $p_ {j}$ in (1) is different from $p_ {j}$ used in Appendix B.**
>
> We have revised this in the updated paper by distinguishing the aggregation weights of $p_ {j}$ and $\hat{p}_ {j}$, corresponding to the aggregation on local updates from all participating clients $\mathcal{J}^{i}$ and subset clients $\hat{\mathcal{J}}^{i}$ in round $i$, respectively.
>
> **Q: I think that $\Theta$ is missing in the probabilistic graphical model depicted in Figure 3.**
>
> We appreciate the reviewer for pointing it out. We have revised Figure 3 to provide a clearer illustration of the graphic model in FedTrans.

---

> ### Author Response · Authors · 2023-11-20
> **Response to reviewer DCr1 (Part 4)**
>
> `[8/10] Could you, please, discuss the fairness implication of FedTrans?`
>
> We appreciate the reviewer’s feedback on pointing out the fairness issue. Please refer to [6/10] for our response on the fairness issue.
>
> `[9/10] Could you, please, clarify if the evaluation reports the performance of benign clients only, or includes the performance of noisy clients?`
>
> We thank the reviewer for the feedback. Please refer to our response in [5/10] for clarification on the evaluation setting.
>
> `[10/10] Could you, please, report the success rate of FedTrans in detecting the noisy clients?`
>
> We have updated Appendix C with experiments on client selection performance, as shown in Figure 10. The FedTrans achieves an approximate 85% success rate in detecting noisy clients when adopting a threshold of 0.5 to exclude the updates from clients with utility $\theta$ below the threshold (the threshold can be further optimized, which we leave for future work as client selection is not our primary focus). We also observe that those noisy clients wrongly recognized as clean clients (i.e., false positives) tend to have a low local noise rate, typically between 0.1 and 0.3. More interpretation of the results is given on page 21, paragraph "Client selection performance".

---

> > ### Comment · Reviewer_DCr1 · 2023-11-22
> >
> > I extend my appreciation to the authors for their insightful rebuttal, which included supplementary experimental results, thereby enhancing the overall quality. Despite these improvements, I maintain that my initial score remains fair, and as such, I refrain from revising it.

---

> ### Author Response · Authors · 2023-11-22
> **Thank you for your reply**
>
> Dear Reviewer DCr1,
>
> Thank you for taking the time to review our rebuttal and for acknowledging the supplementary experimental results we provided. We appreciate your insightful review to enhance the quality of our work.
>
> Best,
> Authors

---

### Meta-Review · Area_Chair_cwJb · 2023-12-08

**Metareview:**

This paper presents a new Bayesian method for server-side robust aggregation in federated learning against heterogeneous and noisy data. The proposed method relies on the existence of a (small) clean/balanced dataset at the server side that is used to estimate client utility. Overall, the method seems to be effective as evidenced by the extensive experiments, and the paper is well-written and well communicated. All reviewers mentioned that they were satisfied with the authors' responses to their questions and recommended the paper to be accepted. Congratulations to the authors.

**Justification For Why Not Higher Score:**

I think quotes from the reviewers is self-sufficient for this justification:

-  "The paper is incremental and does not bring significant novel insights to the field." (Reviewer DCr1)

- "apply existing tool (Bayesian framework) for client contribution evaluation in FL, but did not show any new findings/insights" (Reviewer PLVU).

**Justification For Why Not Lower Score:**

- The problem area is important and hence the method could potentially be impactful.

- The paper is well-written and clearly communicated.

- The proposed method performs well (based on extensive evaluations).

---

### Decision · Program_Chairs · 2024-01-16

Accept (poster)